# Machine learning-driven multifunctional peptide engineering for sustained ocular drug delivery

Henry T. Hsueh[1,2,9], Renee Ti Chou[3,9], Usha Rai[1,4], Wathsala Liyanage[1,4], Yoo Chun Kim[1,4], Matthew B. Appell[1,5], Jahnavi Pejavar[1,2], Kirby T. Leo[1,6], Charlotte Davison[1,2], Patricia Kolodziejski[1,2], Ann Mozzer[1,4], HyeYoung Kwon[1,2], Maanasa Sista[1,7], Nicole M. Anders[8], Avelina Hemingway[8], Sri Vishnu Kiran Rompicharla[1,4], Malia Edwards[4], Ian Pitha[1,4], Justin Hanes[1,2,4,5,6,8], Michael P. Cummings[3] ✉ & Laura M. Ensign[1,2,4,5,6,8] ✉

Sustained drug delivery strategies have many potential benefits for treating a range of diseases, particularly chronic diseases that require treatment for years. For many chronic ocular diseases, patient adherence to eye drop dosing regimens and the need for frequent intraocular injections are significant barriers to effective disease management. Here, we utilize peptide engineering to impart melanin binding properties to peptide-drug conjugates to act as a sustained-release depot in the eye. We develop a super learning-based methodology to engineer multifunctional peptides that efficiently enter cells, bind to melanin, and have low cytotoxicity. When the lead multifunctional peptide (HR97) is conjugated to brimonidine, an intraocular pressure lowering drug that is prescribed for three times per day topical dosing, intraocular pressure reduction is observed for up to 18 days after a single intracameral injection in rabbits. Further, the cumulative intraocular pressure lowering effect increases ~17-fold compared to free brimonidine injection. Engineered multifunctional peptide-drug conjugates are a promising approach for providing sustained therapeutic delivery in the eye and beyond.

In many disease settings, sustained delivery of therapeutic levels of drug can improve treatment efficacy, reduce side effects, and avoid challenges with patient adherence to intensive dosing regimens[1,2]. This is particularly critical in the management of chronic diseases, where long-term adherence to medication usage and clinical monitoring can suffer[3,4]. In the ophthalmic setting, the leading causes of irreversible blindness and low vision are primarily age-related, chronic diseases, such as glaucoma and age-related macular degeneration[5–7]. Recent approvals of devices that provide sustained therapeutic release, such as the Durysta® intracameral implant for continuous delivery of an

¹Center for Nanomedicine at the Wilmer Eye Institute, Johns Hopkins University School of Medicine, Baltimore, MD, USA. ²Department of Chemical & Biomolecular Engineering, Johns Hopkins University, Baltimore, MD, USA. ³Center for Bioinformatics and Computational Biology, University of Maryland, College Park, College Park, MD, USA. ⁴Department of Ophthalmology, Johns Hopkins University School of Medicine, Baltimore, MD, USA. ⁵Department of Pharmacology and Molecular Sciences, Johns Hopkins University, Baltimore, MD, USA. ⁶Department of Biomedical Engineering, Johns Hopkins University, Baltimore, MD, USA. ⁷Department of Biomedical Engineering, Case Western Reserve University, Cleveland, OH, USA. ⁸The Sidney Kimmel Comprehensive Cancer Center at Johns Hopkins University, Baltimore, MD, USA. ⁹These authors contributed equally: Henry T. Hsueh, Renee Ti Chou. ✉e-mail: mcummin1@umd.edu; lensign@jhmi.edu

intraocular pressure (IOP) lowering agent, and the surgically implanted port-delivery system that provides continuous intravitreal delivery of ranibizumab, highlight the importance of these next generation approaches for ocular disease management[8–11]. Conventionally, sustained therapeutic effect is achieved by an injectable or implantable device that controls the release of the therapeutic moiety into the surrounding environment. However, these devices typically require injection through larger gauge needles or a surgery for implantation, with both procedures having associated risks[12–14]. Further, the buildup of excipient material, the need for device removal, and the potential for foreign body reaction can cause further issues[10,15,16].

One approach for circumventing the issues associated with sustained release devices is to impart enhanced retention time and therapeutic effect to drugs upon administration to the eye without the need for an excipient matrix/implant. Binding to melanin, a pigment present within melanosomes in multiple ocular cell types, was previously reported to affect ocular drug biodistribution[17]. Due to the low turnover rate of ocular melanin, a drug that can bind to melanin may accumulate in pigmented eye tissues, leading to drug toxicity or drug sequestration[18,19]. However, with the right balance of melanin-binding affinity and capacity, melanin may act as a sustained-release drug depot in the eye that results in prolonged therapeutic action[20]. Several drugs have been demonstrated to have intrinsic melanin binding properties due to particular physicochemical properties, which in some cases, prolongs the pharmacologic activity in the eye[20–22].

To impart beneficial melanin-binding properties to drugs, one approach is to engineer peptides with high melanin binding that could be conjugated to small molecule drugs through a reducible linker. Thus, the peptide would provide enhanced retention time, while the linker would ensure that drug could be released and exert its therapeutic action in a sustained manner. In addition, there are available databases describing how peptide sequence affects cell-penetration[23,24], and separately cytotoxicity[25], enabling the potential for engineering multifunctional peptides that can be chemically conjugated to drugs. Incorporating multiple functions into one peptide sequence remains challenging, and thus multifunctional peptides are often designed by fusing peptides via a linker, thus forgoing potentially more efficient rational design, or by testing additional properties on peptides with known functions[26–28]. In contrast, machine learning could allow for designing peptide sequences that simultaneously provide multiple desired properties.

Here, we describe the development of engineered peptides informed by machine learning, which have three properties: high binding to melanin, cell-penetration (to enter cells and access melanin in the melanosomes), and low cytotoxicity. As there was no prior information for how peptide sequences affect melanin binding, we experimentally determine the effect of peptide sequence on melanin binding using a microarray. We then apply machine learning-based analyses to identify peptide sequences that display all three desired properties. Importantly, with the Shapley additive explanation (SHAP) analysis[29] of peptide variables, the machine learning model interpretation provides additional insights and reasoning for the multifunctionality of the peptides. As a proof-of-principle, we demonstrate here that an engineered peptide, HR97, can be conjugated to the intraocular pressure (IOP) reducing drug, brimonidine tartrate. A single intracameral (ICM) injection of the HR97-brimonidine conjugate is able to provide sustained IOP reduction in normotensive rabbits compared to ICM injection of an equivalent amount of brimonidine tartrate, or a topical dose of Alphagan® P 0.1% eye drops. Further, the maximum measured change in IOP from baseline (ΔIOP) is increased with ICM injection of the HR97-brimonidine conjugate. We anticipate that engineered peptide-drug conjugates will facilitate the development of implant-free injectables for use in a variety of ophthalmic indications.

## Results

### Development of high throughput melanin binding peptide microarray methodology

To determine how peptide sequence affects melanin binding properties, we adapted a high-throughput flow-based peptide microarray system to characterize melanin binding events (Fig. 1a). Commercially available eumelanin was processed into nanoparticles (mNPs) to prevent sedimentation and provide reproducible surface area available for binding to peptides printed on the substrate surface. The mNPs had an mean size of $200.7 \pm 5.99$ nm and ζ-potential of $-23.7 \pm 1.39$ mV (Fig. 1b, c). The mNPs were further biotinylated (b-mNPs) to facilitate fluorescent labeling with streptavidin DyLight680. The b-mNPs showed slightly larger mean size of $216.0 \pm 14.85$ nm and ζ-potential of $-21.2 \pm 2.15$ mV (Fig. 1b, c), and maintained similar spherical morphology (Supplementary Fig. 1a) and binding to small molecule drugs brimonidine tartrate and sunitinib malate (Supplementary Fig. 1b). The first microarray was printed with 119 peptides to screen flow conditions for the highest fluorescent reporter signal, which identified that the 500 μg/mL of biotinylated mNPs in pH 6.5 PBS buffer at room temperature was optimal (Fig. 1d and Supplementary Fig. 2). We then used the fluorescent reporter signals to construct a melanin binding classification random forest model (Supplementary Data 1). The prediction accuracy was 0.92. The permutation-based variable importance analysis[30] further revealed that the net charge, basic amino acids, and isoelectric point (pI) may contribute to distinguishing melanin binding and non-melanin binding peptides (Fig. 1e).

### Training of the melanin binding regression model

A second larger peptide screen was implemented to generate melanin binding data to use for the additional model generation (Fig. 2a). Specifically, we used the trained random forest model to predict melanin binding for ~630,000 randomly generated peptides, and those classified as melanin binding were selected. A total of 5499 peptides were printed in duplicate, and the fluorescent reporter intensities were reported as the amount of the b-mNPs that bind to the printed peptides on the microarray. Surprisingly, we identified 780 peptides displaying higher levels of fluorescent reporter intensities than any of the 16 peptides described in the literature that bound to human melanoma cells[31] and melanized *C. neoformans*[32], which were previously screened by the phage display technique. Furthermore, there were 758 peptides showing higher fluorescent values than the highest melanin binding peptides (661.5 arb. units) from the 119-peptide microarray, demonstrating the enrichment of melanin binding properties from training the random forest model. Next, the fluorescent reporter intensities values were used as the response variable in training a regression model (Supplementary Data 2). Applying a variable reduction procedure using random forest to eliminate less informative variables from the data set, reduced the number of variables from 1094 to 64 (Supplementary Fig. 3a), and model performance measured by the coefficient of determination ($R^2$) improved from 0.48 to 0.53. A wide array of machine learning models was explored and trained on the variable-reduced data set and were integrated with a super learning (SL) framework that combined various types of base models weighted using a meta-learner. By applying the iterative base model filtering procedure (Fig. 2b), the complexity of the SL was further reduced. To explore other combinations of base models in the SL ensemble, homogeneous base models consisting of models from only one algorithm family were constructed. A nested cross-validation (Fig. 2c) was applied to estimate an unbiased generalization performance. All SL models with base model reduction were selected as the top model in the inner loop cross-validations, and the performance evaluated in the outer loop cross-validation improved to $R^2 = 0.54 \pm 0.01$ (Supplementary Table 1). The reduced SL was selected amongst 31 competitive models (Supplementary Data 3) as the final melanin binding regression model. When training the same set of

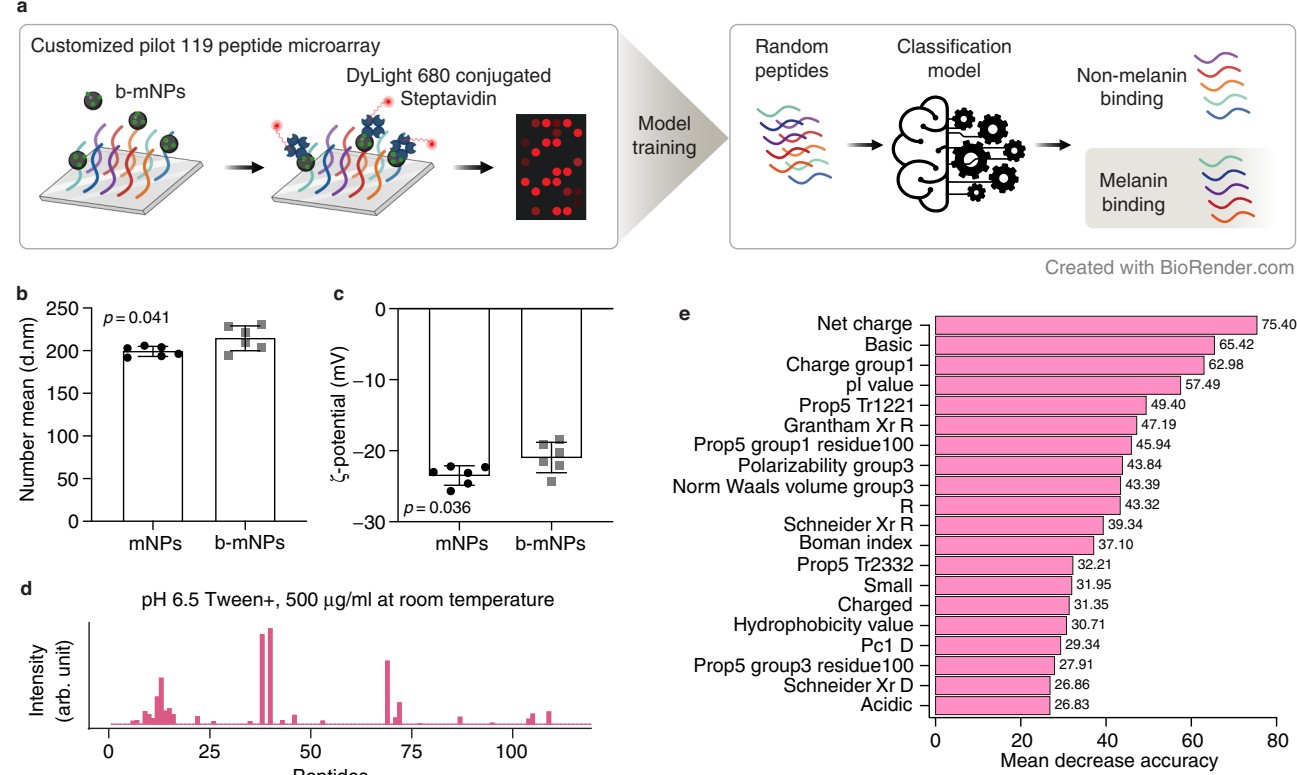

**Fig. 1 | Pilot 119 melanin binding peptide microarray screening with machine learning analysis. a** Schematic illustration of the first peptide microarray. Peptides were anchored to a microarray, and melanin nanoparticles (mNPs) with surface biotinylation (b-mNPs) were flowed over to characterize binding events. The fluorescence intensity of the biotin was detected using DyLight 680-conjugated streptavidin to quantify melanin binding for each peptide. An initial classification model was trained using the data generated. Random peptides were then classified by the model as melanin binding or non-melanin binding. Created with BioRender.com. **b,c** Plot showing the sizes (**b**) and ζ-potential (**c**) of mNPs (black dots,

$n = 6$) and b-mNPs (gray squares, $n = 6$). Data are presented as mean ± SD. Group means were compared using Student's $t$ tests (two-tailed). **d** The optimal interaction profiling of b-mNPs against 16 positive control peptides (peptide numbers: 1–16) and 103 random peptides (peptide numbers: 17–119). **e** Permutation-based variable importance analysis of the melanin binding classification random forest. The $x$-axis indicates the mean decrease in prediction accuracy after variable permutation. The values are shown at the end of the bars. The top 20 important variables ranked by mean decrease in accuracy are shown. See Supplementary Data 8 for detailed variable descriptions.

models on the whole data set, and number of base models in the SL was reduced from 907 to 38 (Fig. 2d). Adversarial computational control was performed, and the generalization performance was $R^2 = -0.04 \pm 0.02$, indicating that the machine learning was effective in learning meaningful relationships in the melanin binding data set.

## Training of cell-penetration and cytotoxicity classification models

Engineered peptides must enter cells to reach and bind to melanin within the melanosomes and should be minimally toxic to cells. Thus, the SkipCPP-Pred[23] and the ToxinPred[25] databases were used to create SL classification ensembles to engineer tri-functional peptides (Supplementary Data 4, 5). Variable reduction decreased the number of variables from 1094 to 11 for the cell-penetration data set (Supplementary Fig. 3b) and from 1094 to 56 for the cytotoxicity data set (Supplementary Fig. 3c). The prediction accuracies calculated from out-of-bag samples improved from 0.91 to 0.93 and from 0.951 to 0.958 for cell-penetration and cytotoxicity, respectively. We subsequently trained base models and SL ensembles, and the generalization performances in terms of Matthews correlation coefficient (MCC), $F_1$ (harmonic mean of precision and recall), and balanced accuracy for cell-penetration were $0.79 \pm 0.01$, $0.90 \pm 0.01$, and $0.90 \pm 0.01$, respectively; and those for cytotoxicity were $0.88 \pm 0.004$, $0.92 \pm 0.002$, and $0.95 \pm 0.002$, respectively (Supplementary Tables 2, 3). The number of base models in the reduced SL models trained on the whole data sets were decreased from 310 to 65 for cell-penetration,

and from 311 to 22 for cytotoxicity (Supplementary Fig. 4). There were 300 competitive cell-penetration models and 175 competitive cytotoxicity models (Supplementary Data 6, 7). A GBM model and the reduced SL were selected as the final predictive cell-penetration and cytotoxicity models. Similar to melanin binding, adversarial controls had decreased generalization performances, where the MCC, $F_1$, and balanced accuracy were $-0.002 \pm 0.05$, $0.52 \pm 0.03$, and $0.50 \pm 0.03$ for cell-penetration, and $0.001 \pm 0.01$, $0.05 \pm 0.02$, and $0.62 \pm 0.04$ for cytotoxicity.

## Validation of predicted peptide properties in vitro

A position-dependent amino acid frequency matrix was used to generate 127 peptides that spanned the range of low to high predicted melanin binding. Among the 127 peptide candidates, 113 peptides were classified as cell-penetrating and 117 peptides were predicted as non-toxic. To experimentally measure melanin binding in vitro, biotinylated peptides were incubated with mNPs, and the bound fraction was calculated using an avidin-based fluorescent reporter (Fig. 3a). The Pearson correlation coefficient was computed to compare the predicted and experimental melanin binding values, and the correlation coefficient was $r = 0.84$, showing a high level of correlation between the predicted and experimental values (Fig. 3b). We next characterized how the predicted cell-penetrating properties of the peptides affected cell uptake in a retinal pigment epithelium cell line (ARPE-19). ARPE-19 cells were cultured using standard methods (non-induced, $n = 3$) and using culture conditions that induce melanin production (induced,

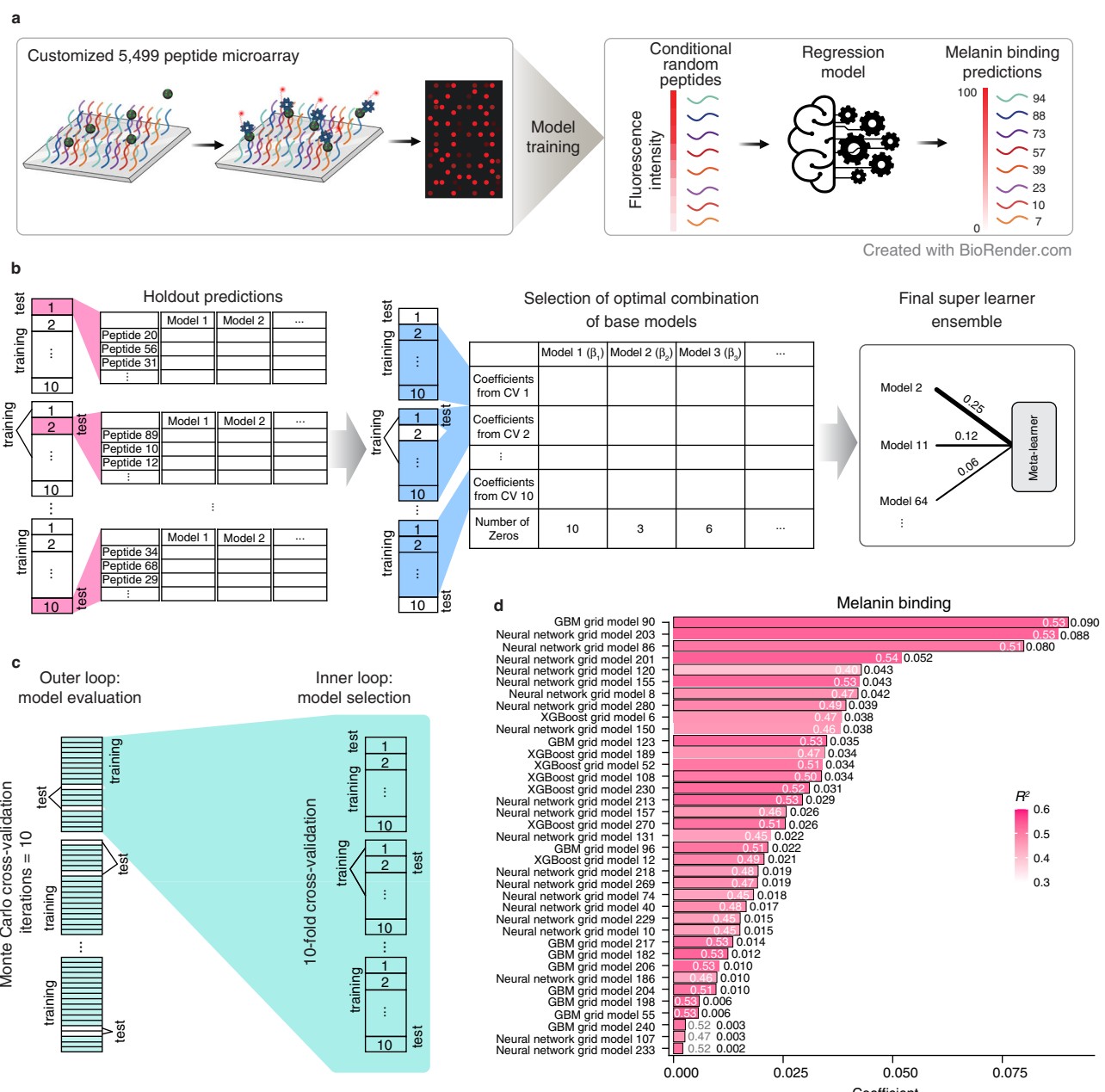

**Fig. 2 | Schematic of the machine learning pipeline based on the super learner framework for the melanin binding data set. a** Scheme of a larger microarray, which includes 5499 peptides used to train a regression super learner. Random peptides were generated based on position-dependent amino acid frequencies calculated using the second peptide array data, and the melanin binding levels were predicted. Peptides with desired melanin binding levels were selected for further experimental validation. Created with BioRender.com. **b** Scheme of the super learner complexity reduction. Holdout predictions of peptides (shown as rows) were generated for each base model (shown as columns) with tenfold cross-validation (CV) on the input data set. A meta-learner (generalized linear model) was fitted on the holdout predictions with another tenfold cross-validation. The number of base models was reduced by applying an iterative reduction procedure (see

*Methods*). The final super learner ensemble was trained on the input data set with the optimal combination of the selected base models. **c** Scheme of the machine learning pipeline for an unbiased model performance evaluation. The nested cross-validation includes an outer loop for model evaluation and an inner loop for model selection (cyan). The outer loop generated 10 sets of train-test splits using a Monte Carlo method, and the inner loop generated 10 sets of train-test splits using a modulo method. **d** Plot of the base models of the final melanin binding super learner. Coefficients of determination ($R^2$) are denoted with color and conveyed as white text on the bars or gray text adjacent bars. Base model coefficients are indicated at the bar ends. There is one model having zero coefficient and not shown. See *Methods* and Supplementary Note 2 for information about model hyperparameter details and statistics of model performance.

$n = 3$[20]. A positive correlation was observed between the measured in vitro melanin binding of the peptides and the intracellular peptide concentrations in melanin-induced cells for cell-penetrating peptides ($r = 0.77$, $p < 2.2 \times 10^{-16}$, Fig. 3c, d), suggesting correlation between the two properties. Further, peptides predicted to be cell-penetrating demonstrated significantly

higher intracellular concentrations (median 229.4 pmol/100 K cells) than those of non-cell-penetrating peptides (median 26.7 pmol/100 K cells) in the melanin-induced cells ($p = 6.9 \times 10^{-6}$, Fig. 3e). In contrast, the intracellular peptide concentrations were not affected by the predicted properties in non-induced cells (Supplementary Fig. 5).

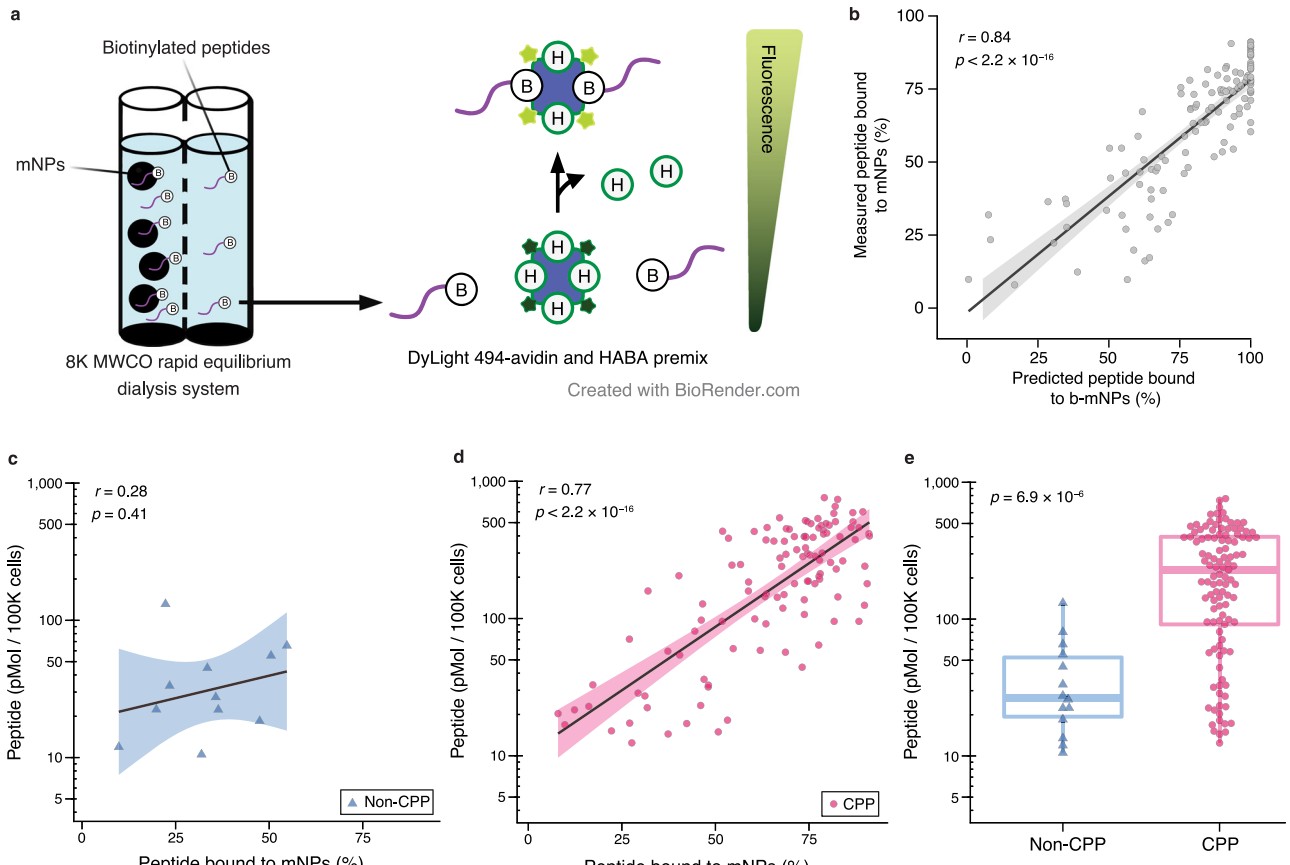

**Fig. 3 | Experimental validations of final model predictions on melanin binding and cell-penetration. a** Schematic showing an in vitro melanin binding assay with melanin nanoparticles (mNPs) using a biotin quantification kit. The DyLight 494-tagged avidin emitted fluorescence when the biotinylated peptides displaced the weakly interacting 4'-hydroxyazobenzene-2-carboxylic acid (HABA or H). Created with BioRender.com. **b** Plot of the relationship between predicted melanin binding and binding measured experimentally in vitro. The x-axis indicates melanin binding predictions from the final super learner, and the y-axis indicates the experimental melanin binding values ($n = 4$ for each peptide). Dots represent the mean value for peptides. The black linear trend line conveys the Pearson correlation relationship (two-tailed), and the gray area indicates the 95% confidence interval.
**c, d** Comparison of melanin binding and cell-penetration in melanin-induced

human adult retinal pigment epithelial (ARPE-19) cells. Blue triangles denote predicted non-cell-penetrating peptides (non-CPP), and magenta dots represent predicted cell-penetrating peptides (CPP). The x-axes indicate melanin binding measured in vitro ($n = 4$ for each peptide), and the y-axes convey intracellular peptide concentration measured from the cell uptake assay ($n = 3$ for each peptide). Black linear trend lines indicate Pearson correlation relationships, with 95% confidence intervals shown as shaded areas. The correlation coefficients and p-values (two-tailed) are shown. **e** Summary of CPP ($n = 113$) and non-CPP ($n = 14$) intracellular concentrations. Box plot conveys median (middle line), 25th and 75th percentiles (box), and the $1.5 \times$ interquartile range (whiskers). The p value was calculated using a Mann–Whitney U test (two-tailed).

## Analysis of peptide variables that contribute to observed properties

To identify which peptide variables contributed to the properties observed in vitro, Shapley additive explanation (SHAP) analysis of the final predictive models was performed. The results showed that peptide property predictions were based on contribution from multiple variables. More specifically, basic peptides and higher net charge variables had higher contributions to melanin binding predictions (Fig. 4a), which was consistent with the top variables identified by the random forest classification model trained on the pilot peptide microarray. Similarly, higher net charge and higher isoelectric point contributed more to cell-penetration (Fig. 4b), and less free cysteines had more influence on non-toxic predictions (Supplementary Fig. 6). To understand how reliable the interpretable results were, adversarial controls were constructed with the final predictive models using a 10-fold cross-validation. Indeed, the distributions and levels of variable contributions changed for melanin binding, cell-penetration, and cytotoxicity (Supplementary Fig. 7). Among all the peptide candidates, HR97 (FSGKRRKRKPR) was selected based on combination of

the three peptide properties (melanin binding$_{HR97}$ = 79.1 ± 0.7%, cell uptake$_{HR97}$ = 759.9 ± 19.6 pmol/100 K cells, non-toxic$_{HR97}$ = 96.9%, Fig. 4c). HR97 had the highest intracellular concentration, which outperformed the well-characterized cell-penetrating peptide fragment of the HIV trans-activator protein (TAT$_{47-57}$, YGRKKRRQRRR). HR97 demonstrated increased cell uptake compared to TAT$_{47-57}$ in both the induced ARPE-19 cells (cell uptake$_{HR97}$ = 759.9 ± 19.6 pmol/100 K cells, cell uptake$_{TAT47-57}$ = 457.1 ± 34.2 pmol/100 K cells) and the non-induced cell type (cell uptake$_{HR97}$ = 82.5 ± 9.1 pmol/100 K cells, cell uptake$_{TAT47-57}$ = 68.3 ± 4.6 pmol/100 K cells). In addition, HR97 showed no sign of cytotoxicity in ARPE-19 cells at concentrations up to 5 mg/mL (Supplementary Fig. 8). HR97 predictions embodied all the properties that were the largest contributors to each functionality, including being basic (63.64% basic amino acids), possessing a high net charge (6.98) and a high isoelectric point value (12.99), and no cysteines (Fig. 4d–f). By visualizing the peptide design space defined by the sequences and variables used in training the desired functional properties, the peptide candidates with high melanin binding predictions were shown up in the same cluster, showing similar sequence

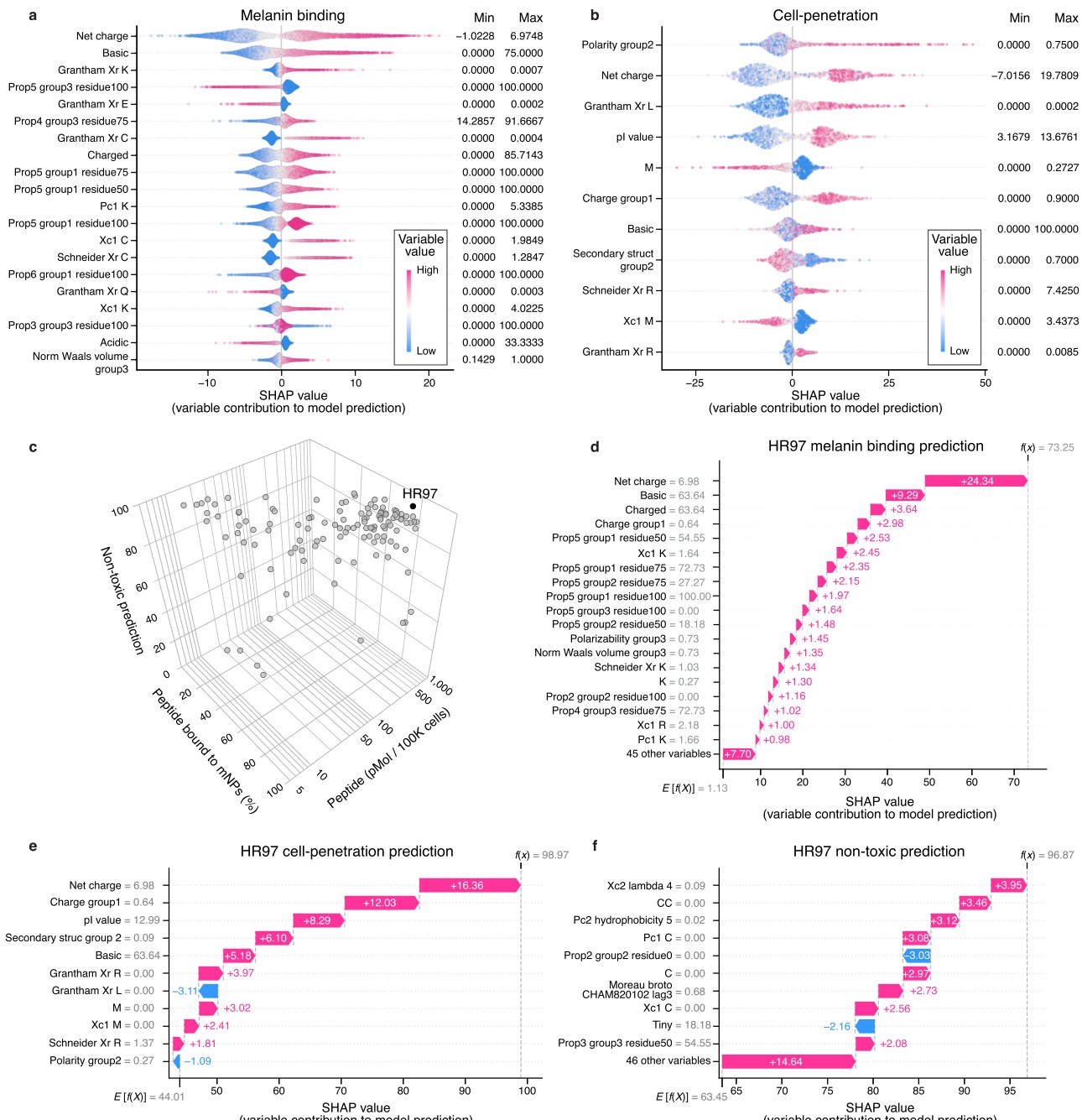

**Fig. 4 | Melanin binding, cell-penetration model interpretation, and variable contributions to HR97 multifunctional peptide predictions.** Overall variable contributions to model predictions for (**a**) melanin binding and (**b**) cell-penetration. The top important variables analyzed using Shapley additive explanations (SHAP) are shown. Dots represent peptides from cross-validation test sets. The x-axes indicate SHAP values, indicative of variable contributions to model prediction ranging from 0 to 100. The variables were ranked based on the difference between the maximum and minimum SHAP values. The color gradient indicates the variable values normalized by percentile ranks. Higher variable values are indicated by darker magenta color and lower values by darker blue color. The minimum and maximum variable values are noted on the right of each subplot. **c** Scatter plot showing the in vitro melanin binding, in vitro cell-penetration, and predicted cytotoxicity values of the 127 candidate peptides. Dots represent peptides. HR97 (black dot) was selected based on the optimal multifunctional combination. **d**–**f** Variable contributions to HR97 multifunctional predictions for melanin binding, cell-penetration, and cytotoxicity. The top variables ranked by absolute SHAP values are shown. Magenta bars indicate positive contributions, and blue bars are negative contributions. The y-axis labels convey variable names and their values for HR97. E[f(X)] denotes the expected prediction value, and f(x) is the final prediction, calculated from the sum of all SHAP values plus E[f(X)]. See Supplementary Data 8 for detailed variable descriptions.

motifs and physiochemical properties (Fig. 5a, b). Further, peptides predicted to have high melanin binding were mostly predicted to be cell-penetrating, but cell-penetrating peptides may not be melanin binding (Fig. 5c). The results also suggest that some melanin binding peptides may be toxic (Fig. 5d).

## Characterization and validation of a peptide-drug conjugate in vivo

To investigate the effect of peptide conjugation on drug pharmacodynamics, we chose brimonidine tartrate, a topical IOP lowering drug prescribed for glaucoma treatment. The HR97 peptide was conjugated

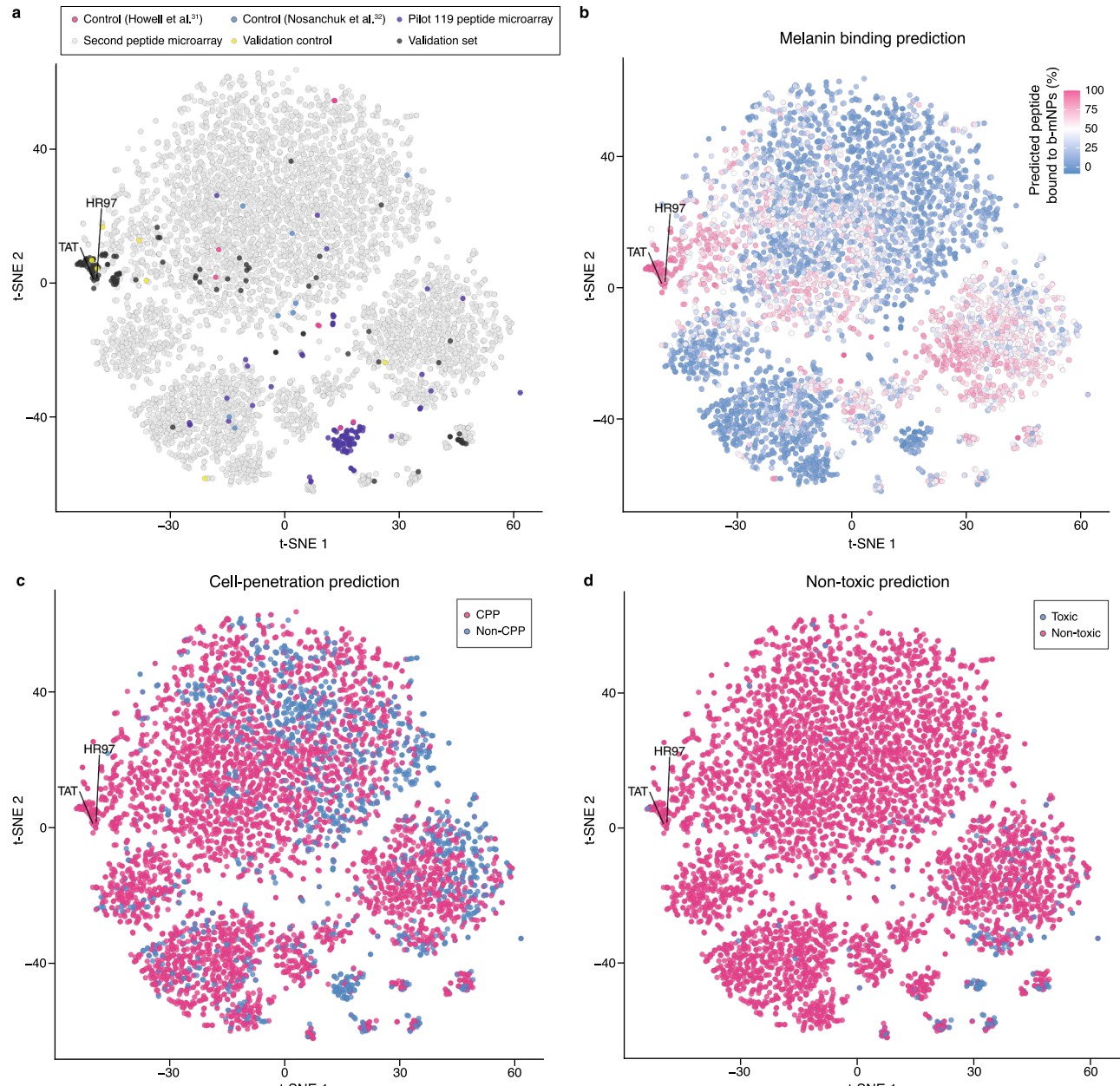

**Fig. 5 | Visualization of the peptide design space based on sequences and physiochemical properties. a** t-distributed stochastic neighbor embedding (t-SNE, used to visualize high-dimensional data) plots showing the peptide design space defined by the combination of one-hot encoded peptide sequences and variables used in melanin binding, cell-penetration, and cytotoxicity model training. Dots represent control peptides from Howell et al.[31] (magenta) and Nosanchuk et al.[32] (blue); peptides used in the pilot (purple) and second (gray and yellow) melanin binding peptide microarrays; and multifunctional peptide candidates (black and yellow) used in the validation experiments. HR97 and TAT are noted. **b** t-SNE plot of peptides colored by melanin binding prediction. Higher melanin binding values are colored by darker magenta and lower by darker blue. **c** t-SNE plot of peptides colored by cell-penetration prediction. Magenta dots represent predicted cell-penetrating peptides (CPP), and blue dots are predicted non-cell-penetrating peptides (non-CPP). **d** t-SNE plot of peptides colored by cytotoxicity prediction. Blue dots denote predicted toxic peptides, and magenta dots indicate non-toxic peptides.

to brimonidine (HR97-brimonidine) via a quaternary-ammonium traceless linker system, and the structure of the intermediates and the purified conjugate were validated by NMR and MALDI-TOF (Supplementary Figs. 9–12). Conjugation to HR97 provided a ~10-fold increase in the in vitro melanin binding capacity of brimonidine ($5.9 \times 10^{-7}$ $K_d$ (M) vs. $5.0 \times 10^{-8}$ $K_d$ (M)), which brought the binding capacity closer to other drugs with high intrinsic melanin binding, such as sunitinib malate (Fig. 6a)[20,33–37]. When incubated in human aqueous fluid, only ~7% of the brimonidine was released from the HR97-brimonidine conjugate over 28 days in vitro (Fig. 6b). However, upon incubation with supraphysiological concentrations of human cathepsin cocktails

to enzymatically cleave the linker, ~52% of the brimonidine was liberated within 48 h (Fig. 6c). The effect of the HR97-brimonidine conjugate on IOP was then evaluated in normotensive Dutch Belted rabbits. A single topical dose with the commercial brimonidine eye drop ($n = 5$) was found to provide a peak reduction in IOP from baseline (ΔIOP) of $-3.0 \pm 0.82$ mmHg that recovered to baseline within 8 h (Fig. 6d). In contrast, a single ICM injection of the HR97-brimonidine conjugate resulted in a greater peak ΔIOP compared to an ICM injection of brimonidine solution at 2 days ($-4.9 \pm 0.46$ mmHg vs. $-2.6 \pm 1.65$ mmHg, $p < 0.05$, red arrow). In a separate experiment, ICM injection of saline or HR97 ($n = 5$ for each) resulted in a similar decrease

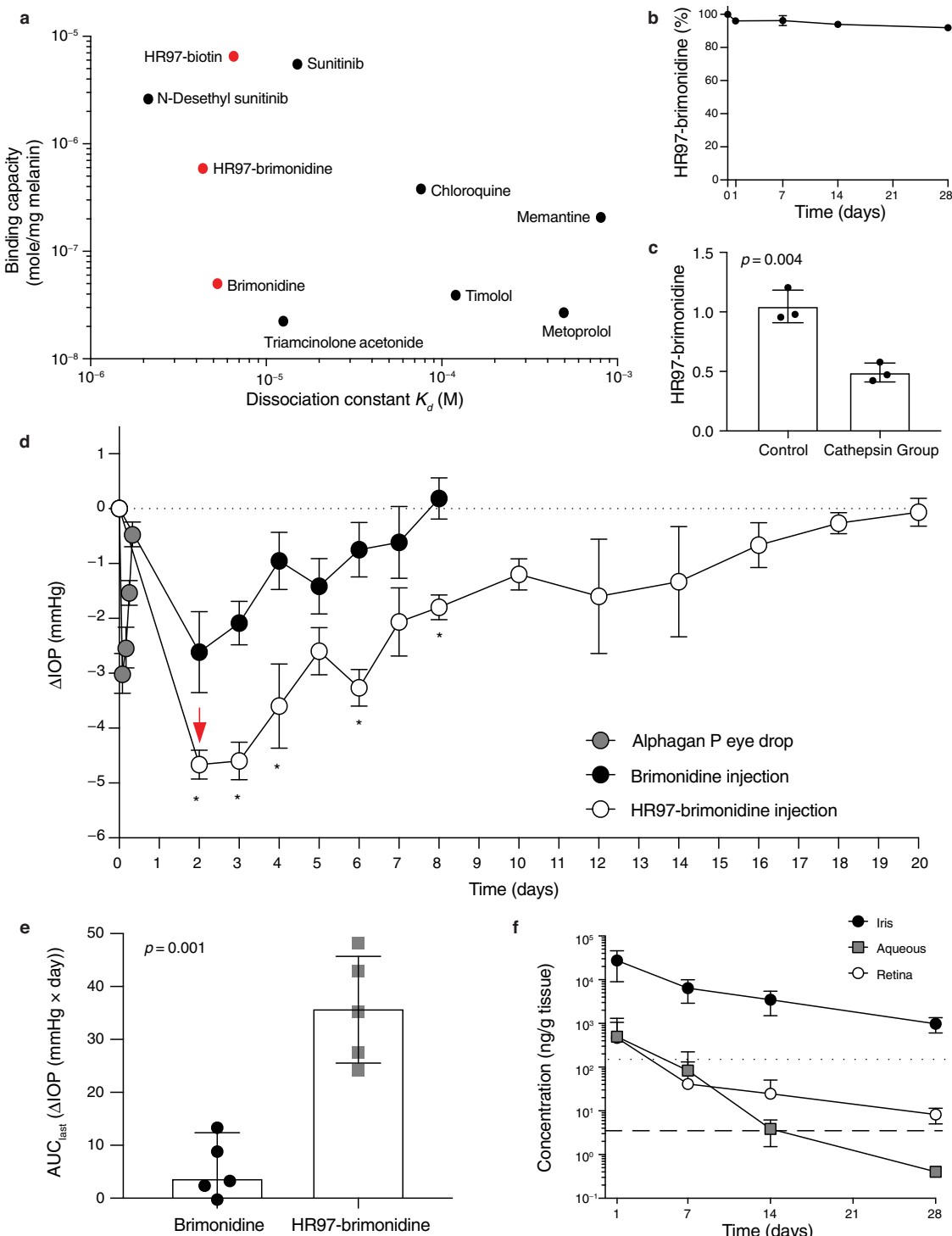

**Fig. 6 | Characterization of HR97-brimonidine in vitro and in vivo. a** In vitro binding capacity and dissociation constant of HR97-biotin, HR97-brimonidine, and brimonidine characterized using a melanin nanoparticle (mNP) assay (red dots, $n = 3$–5). Values shown for comparison include those we previously measured for sunitinib and N-desethyl sunitinib[20], and literature values for other ophthalmic drugs[33–37]. **b** In vitro stability of HR97-brimonidine conjugate in human aqueous humor for 28 days. The percent remaining was normalized to the starting concentration on day 0 ($n = 3$). Data are shown as mean ± SD. **c** Cathepsin cleavage assay of the HR97-brimonidine conjugate. HR97-brimonidine ($n = 3$) were incubated with human cathepsin cocktails or buffer only for 48 h at 37 °C (two-tailed $t$-test). Data are shown as mean ± SD. **d** Comparison of the intraocular pressure (IOP) change from baseline (ΔIOP) after a single ICM injection of HR97-brimonidine conjugate (white dots), brimonidine solution (black dots, 200 μg brimonidine equivalent), and a single drop of Alphagan P (gray dots, 0.15%) in normotensive

Dutch Belted rabbits ($n = 5$ per group). The IOP was measured every 1–2 days until returning to the baseline. The red arrow highlights the further decrease in IOP provided by the HR97-brimonidine. Two-tailed $t$-test was used, *$p < 0.05$ (adjusted $p$ values for days 2, 3, 4, 6, and 8 were 0.044, 0.007, 0.038, 0.007, 0.007, respectively). Data are presented as mean ± SEM. **e** Cumulative ΔIOP of brimonidine (black dots) and HR97-brimonidine (gray squares) after ICM injection. The cumulative ΔIOP was characterized by calculating the area under the curve over the 20-day measurement period (AUC$_{last}$, $n = 5$). Two-tailed $t$-test was used. Data are presented as mean ± SD. **f** Levels of brimonidine in the iris (black dots), aqueous (gray squares), and retina (white dots, $n = 3$–4) over time after ICM injection of HR97-brimonidine (200 μg brimonidine equivalent). The concentrations of brimonidine measured in the aqueous after a single drop of Alphagan P (0.15%) as part of a previous study[38] at 2 h (maximal IOP lowering time point; dotted line) and 4 h (dashed line) after dosage are shown. Data are shown as mean ± SD.

in IOP that returned to baseline by day 3, and ICM injection of a physical mixture of HR97 and brimonidine tartrate ($n = 5$) resulted in a similar IOP profile to the brimonidine solution, returning to baseline by day 8 (Supplementary Fig. 13). To ensure that the dramatic decrease in IOP with the HR97-brimonidine conjugate was not due to toxicity, a board-certified ophthalmologist evaluated the eyes injected with the HR97-brimonidine conjugate on day 7. It was observed that the lids, lashes, and conjunctiva were normal, the corneas were clear, the corneal endothelium was normal without any pigment deposition, the anterior chambers were normal depth, there was no apparent inflammation or fibrin strands, the lenses were clear, and the iris pigmentation was symmetric. According to the same evaluation methods, no ocular toxicity was observed upon ICM injection of saline, HR97, or a physical mixture of HR97 and brimonidine tartrate for at least 28 days (Supplementary Tables 4–7). The mean ΔIOP in the HR97-brimonidine conjugate group remained significantly larger than in the rabbits dosed with brimonidine solution or the physical mixture of HR97 and brimonidine tartrate for up to 14 days (Fig. 6d, Supplementary Fig. 13). Further, the time for the mean ΔIOP to return to baseline was 20 days in the HR97-brimonidine conjugate group compared to 8 days in both groups of rabbits dosed with brimonidine solution or the physical mixture of HR97 and brimonidine tartrate. When summing the area under the curve ($AUC_{last}$) for the cumulative ΔIOP over the 20-day measurement period after ICM injection, the HR97-brimonidine conjugate showed a ~17-fold greater AUC compared to brimonidine solution ($p < 0.001$) (Fig. 6e). A pharmacokinetic study was conducted separately to characterize the intraocular distribution of brimonidine after ICM injection of HR97-brimonidine in Dutch Belted rabbits. The brimonidine concentration remained relatively high in the pigmented iris tissue (980 ng/g) compared to less pigmented parts of the eye, such as the aqueous humor (0.4 ng/g) and the retina (8.3 ng/g) up to 28 days after a single ICM injection (Fig. 6f). The brimonidine concentration in the aqueous on day 7 (83.3 ng/g) was similar to what we previously reported at 2 h after a single drop of Alphagan P (0.15%) (105 ng/g), which was the time with the largest IOP reduction in that study[38]. On day 14 after ICM injection of HR97-brimonidine, the brimonidine concentration in the aqueous (3.9 ng/g) was similar to what we previously reported at 4 h after a single drop of Alphagan P (0.15%) (4 ng/g)[38].

## Discussion

Chronic eye diseases such as glaucoma require continuous treatment to prevent disease progression. Eye drops are the most common dosage form of glaucoma therapy, though low adherence to intensive drop dosage schedules is a major challenge in disease management[3,39,40]. One study using an electronic monitoring device found that only 64% of patients adhered to the three-times daily dosing schedule for brimonidine eye drops over a 4-week period, even though they were aware of the monitoring[41]. Sustained drug delivery systems may be an attractive alternative for the management of chronic ocular diseases like glaucoma. The first sustained-release polymer-based implant for glaucoma treatment, Durysta®, was recently approved for sustained IOP lowering for several months with a single ICM injection[9]. However, the polymer matrix typically took longer to biodegrade than the duration of drug release, and repeated injection with additional implants was associated with increased risk of corneal endothelial cell loss and other corneal adverse reactions[42]. In contrast to conventional polymer-based sustained drug delivery systems, the approach we describe here does not require an implant or large amounts of excipients that will remain in the eye for extended periods. By utilizing short peptide sequences that impart melanin binding to the drug conjugate, a sustained intraocular drug release system was created without the need for a polymer matrix.

Ocular melanin is a biopolymer that resides within melanosomes in pigmented ocular tissues, including the iris, ciliary body, choroid, and retinal pigment epithelium (RPE)[43]. Although the amount of pheomelanin in the eye varies depending on eye color, the amount of eumelanin in ocular tissues, including the RPE, iris pigment epithelium, and pigmented ciliary epithelium is more consistent across the population[44]. It has been described that drug binding to melanin and accumulation inside cells may diminish therapeutic effect by sequestering the drug or causing ocular toxicity[18,19]. In the case of atropine, the intrinsic melanin binding properties were shown to lead to prolonged residence time in pigmented rabbit ocular tissues[45], and a sustained miotic response in pigmented rabbits[21]. In addition, we previously demonstrated that improving the intraocular absorption of sunitinib, a drug with relatively high melanin binding capacity, with a novel gel-forming hypotonic eye drop led to prolonged therapeutic effect of up to 1 week after dosing[20]. Indeed, a recent study used machine learning methods to characterize the structural features of small molecule drugs that impact intrinsic melanin binding, leading to the development of a model that predicted intrinsic melanin binding with 91% accuracy[22]. These findings motivated us to develop engineered adaptors designed to impart tunable melanin binding properties to small molecule drugs used to treat ocular diseases. Further, as melanin is contained within cells, the engineered adaptor should additionally provide cell penetration. Here, we developed a machine learning-based methodology to engineer tri-functional peptides that displayed melanin binding, cell-penetration, and non-toxic properties. The peptide sequence that provided the optimal combination of high melanin binding, high cell-penetration, and low cytotoxicity, HR97, was then conjugated to brimonidine as a proof-of-principle. The HR97-brimonidine conjugate provided up to 18 days of IOP lowering with a single ICM injection in normotensive rabbits, which contrasts with the 8 h-effect provided by a brimonidine eye drop.

Peptides are short sequences of amino acids that can have many combinations with diverse biological functions. Compared to other aptamers and small molecule drug libraries, peptides are relatively cost effective to synthesize and are relatively easy to modify or conjugate to small molecule drugs[46]. Currently, there are more than 80 FDA-approved peptide drugs and more than 600 in clinical and preclinal trials[47–49]. Peptides optimized for a single function, either exhibiting cell-penetration or cell targeting properties, have been widely exploited as drug carriers to shuttle drugs across biological barriers[50–52]. Peptides such as TAT, penetratin, PEP-1 and polyarginine (R6 or R8) and have been conjugated with various cargos for targeting the anterior and posterior segment[53–60]. For example, various fluorescein conjugated peptides were screened for the ability to cross porcine cornea ex vivo[60,61]. Penetratin (PNT) showed an eightfold increase in permeability compared to PEP-1, though most of the peptide was found to be sequestered within cells rather than having crossed the cornea[60,61]. In another study, TAT peptide was conjugated to human acidic fibroblast growth factor (aFGF) and applied topically to rat eyes[62]. They found that the conjugates reached the retina with a $t_{max}$ of 30–60 min and with possible mechanism of conjunctival-scleral penetration route[62]. However, it is known that drugs can more easily reach the posterior segment with topical administration in rat and mouse eyes compared to larger eyes, such as rabbits[63–65].

Many peptide screening technologies have been developed for identifying novel functional peptides, including phage display, mRNA display, and peptide microarray[66–68]. Phage display and mRNA display are capable of screening a larger number of peptides (~$10^{11}$–$10^{13}$) compared to peptide microarray (~$10^5$). However, in phage display and mRNA display, the peptide sequences are randomly generated with fixed ratios of amino acids[67]. In contrast, coupling computationally generated peptide sequences with peptide microarrays has the advantage of rapidly improving peptide design through machine learning model refinement. Peptides can be computationally represented by physicochemical and structural descriptors[69] or encoded using various rules such as binary encoding and evolution-based

encoding[70]. Since peptide sequence is the source of functionality, a machine learning-based approach can be employed to develop predictors that learn the relationships between peptide variables derived from the sequence and the desired functional property[71–73]. Peptide databases have also been made available for data-driven functional peptide design, including cell-penetration and toxicity[24,25]. However, there is only a limited number of studies for, and no database of, melanin binding peptides. An example here is that in the two studies that reported peptide sequences that were characterized as melanin binding, phage display was used to identify 8 peptides that bind to melanin in human melanoma cells[31] and 8 peptides that bound to melanized *C. neoformans*[32]. However, in our peptide microarray, 8 of these peptides did not demonstrate detectable melanin binding, and overall, we identified 780 peptides displaying higher levels of melanin binding than any of these peptides described in the literature. Furthermore, the second peptide microarray designed using the initial machine learning model provided more potent melanin binding peptides compared to the first peptide microarray, demonstrating the rapid improvement in design by machine learning model refinement.

Multifunctional peptides with dual or triple pharmacological properties have also been integrated into drug delivery systems through conjugation to drugs or drug-loaded cargos[26,74,75]. However, it is challenging to design peptides with multiple functions contained in a single sequence. Often single function peptides are fused directly or by a linker peptide[75–77], which may increase the peptide length and reduce the desired functional properties of each component. Another approach is to optimize additional functional properties by substituting amino acids on a template peptide with a known function[27,28], which may require extensive laboratory screening and is time-consuming. Generating multifunctional peptides with the flexibility to choose the desired functional levels is a less explored research area[78,79]. Here, our machine learning and model interpretation approach guided the engineering of multifunctional peptides. The peptide properties were analyzed using the shared variable set, revealing mutually important variables contributing to both melanin binding and cell-penetration, where peptides with moderate to high net charge and containing more basic amino acids tend to possess both melanin binding and cell-penetrating properties. Further, we unexpectedly observed correlation between melanin binding and cell-penetrating in cell uptake in vitro. Thus, the highest intracellular accumulation was achieved by increasing the amount of peptide that can access intracellular melanosomes, where the peptides can then bind to melanin and provide sustained drug release.

Many machine learning models including random forest, support vector machines, and deep learning have been developed to predict how amino acid sequence governs peptide properties[80]. Super learning is an ensemble machine learning method that takes advantage of various machine learning models. The predictive performance of a super learner ensemble is assured to be at least as accurate as the best-performing base model[81,82]. The same model types with varying hyperparameter combinations can be included in a SL ensemble. Recently, it was described that base model hyperparameter tuning could improve overall SL model performance[83]. Based on this finding, we further developed a procedure to systematically select optimal base model composition by iteratively filtering out models that have less contributions to the SL ensemble. Indeed, we obtained better SL model performance compared to the one including all base models. In this study, we explored a wide array of possible machine learning models and identified multiple competitive models through statistical analyses. SL provided a framework to integrate these explored models. Although the meta-learner may add a layer of complexity, it demonstrated an interpretable summary of the model importance in terms of their contributions to the final predictions. In addition, the complexity of the machine learning architecture was reduced by variable reduction of the data sets and base model filtration of SL. Further,

interpretable machine learning that extracts relevant information such as variable contributions to output predictions from the data relationships learned by the model is important for explaining model predictions[84,85]. Many of the functional peptide predictors and other drug discovery tools do not have information on how and why top candidates were identified[86–88]. In this study, we showed that interpretation of machine learning models can provide insights to improve the design of multifunctional peptides. The SHAP analysis not only indicated important variables contributing most to the model prediction, but also showed the relationships between variable values and prediction outputs.

The studies described here are not without limitations. First, while the in vitro ARPE19 cell assay helped validate the cell-penetrating and melanin binding performance, the methodology used here did not differentiate between peptides that were free or bound to melanin or other structures within the cell. Indeed, there was a baseline level of peptide associated with non-pigmented cells, but a substantial increase in cellular localization was observed when the cells were induced to produce melanin. Second, the traceless linker conjugation yield of the HR97-brimonidine was low and requires further optimization. The cathepsin-labile linker was chosen because cathepsins are largely located intracellularly and are present in minimal amounts in extracellular fluids such as aqueous humor[89–92]. Thus, the intracamerally delivered HR97-brimonidine would be stable until it had localized within melanin-containing cells. However, the level of brimonidine measured in rabbit iris tissue remained high, suggesting that further optimization of the linker cleavage and brimonidine release rate may also extend the duration of the therapeutic effect. Finally, the duration of IOP lowering reported here (20 days) was sufficient to demonstrate the proof-of-principle in normotensive rabbits but would not be clinically translatable. Future work with more potent drugs may increase the duration of action.

The approach we described here to apply ensemble machine learning to peptide microarray enabled the efficient design of multifunctional peptides, which in this application enhanced the intraocular pharmacokinetics and pharmacodynamics of the ophthalmic drug brimonidine. Engineered HR97 peptide demonstrated increased cell-penetrating properties compared to known cell-penetrating peptides, such as TAT, and simultaneously possessed high melanin binding capacity and low cytotoxicity. In the current context, utilizing short peptide sequences that impart melanin binding to a drug conjugate may provide an avenue for creating safe and effective implant-free sustained intraocular drug release systems. More broadly, the approach described here can be applied to generate multifunctional peptide-drug conjugates for a variety of biomedical applications.

## Methods

### Material sources
Brimonidine was purchased from TCI America. Eumelanin from *Sepia officinalis*, 0.22 µm Millex-GV PVDF filter, ferric ammonium citrate, bovine serum albumin (BSA), Tween 20, fetal bovine serum (FBS), trifluoroacetic acid (TFA), tert-Butyl methyl ether (MTBE), thionyl chloride, Tetrabutylammonium iodide, N,N-diisopropylethylamine, human cathepsins B, K, L and S, Whatman® Anotop® 0.02 µm syringe filter and Triton X-100 were purchased from Sigma Aldrich (St. Louis, MO, USA). ARPE-19 (ATCC CRL-2302, lot No. 70013110), and DMEM:F12 medium were purchased from the American Type Culture Collection (Manassas, VA, USA). EZ-Link™ Amine-PEG$_2$-Biotin, BupH MES buffer saline pack (2-(N-morpholino)ethanesulfonic acid buffer), EDC (1-ethyl-3-(3-dimethylaminopropyl)carbodiimide hydrochloride), NHS (N-hydroxysuccinimide), Pierce™ Fluorescence Biotin Quantitation Kit, rapid equilibrium dialysis (RED) 8 K device, PrestoBlue™ HS Cell Viability Reagent, DMEM with high glucose and pyruvate, Trypsin-EDTA (0.25%) with phenol, RIPA lysis buffer, Streptavidin DyLight 680, and penicillin/streptomycin were purchased from Thermo Fisher

Scientific (Waltham, MA, USA). Disposable PD-10 desalting columns were purchased from VWR. Dulbecco's Phosphate Buffered Saline (DPBS), 1×phosphate buffered saline (PBS), 10×PBS, high-performance liquid chromatography (HPLC) grade acetonitrile, dimethylformamide (DMF), and water were purchased from Fisher Scientific (Hampton, NH, USA). Mc-Val-Cit-PAB was purchased from Cayman Chemical (Ann Arbor, MI, USA). Endotoxin-Free Ultra-pure Water were purchased from MilliporeSigma (Burlington, MA, USA). A Hamilton 1700 Series gas tight syringes (25 μL, Model 1702 RN, 27 gauge) was purchased from Hamilton Company (Reno, NV, USA). BD 1 mL TB syringe with 28 G needles were purchased from BD (San Jose, CA, USA). Isoflurane was purchased from Baxter (Deerfield, IL, USA). Reverse-action forceps were purchased from World Precision Instruments (Sarasota, FL, USA). Neomycin, polymyxin b, and bacitracin zinc ophthalmic ointment was purchased from Akorn (Lake Forest, IL, USA).

## Melanin nanoparticle synthesis and characterization

Melanin nanoparticles (mNPs) were synthesized from the eumelanin of *Sepia officinalis*. In brief, 10 mg/mL of eumelanin was suspended in the DPBS using an ultrasonic probe sonicator (Sonics, Vibra Cell VCX-750 with model CV334 probe, Newtown, CT, USA) by pulsing 1 s on/off at 40% amplitude for 30 min in a 4 °C water bath. The suspension was then filtered through a 0.22 μm Millex-GV PVDF filter and transferred to PD-10 desalting columns. The resulting mNPs solution was lyophilized for 7 days and stored at −20 °C until further use. For mNP biotinylation (b-mNPs), mNPs were suspended in 2 mL MES buffer with 2.4 mg of EDC and 3.6 mg of NHS for 15 min at room temperature to first activate the carboxylic acid groups. To increase the buffer pH above pH 7.4 for amine reaction, 400 μL of 10 × PBS was directly added to the mixture and incubated for 5 min. Various amounts of EZ-Link™ Amine-PEG$_2$-Biotin (5, 15, 20, 30 mg) were reacted with activated mNPs for 2 or 6 h at room temperature. Since all conditions led to a similar degree of mNP biotinylation, reaction conditions using 5 mg of amine-PEG$_2$-biotin with 2 h incubation at room temperature was used moving forward. The reaction mixture was then transferred to PD-10 desalting columns to further collect the b-mNPs. To transfer the b-mNPs to different solvents (water, pH 6.5 PBS, pH 7.4 PBS) for optimization of the peptide microarray, PD10 columns were first equilibrated with buffer, and then the b-mNPs were added. Particle size and ζ-potential were determined by dynamic light scattering and laser Doppler anemometry, respectively, using a Zetasizer Nano ZS90 (Malvern Instruments). Size measurements were performed at 25 °C at a scattering angle of 173°. Samples were diluted in 10 mM NaCl solution (pH 7), and measurements were performed according to instrument instructions. Pierce™ Fluorescence Biotin Quantitation Kits were used to quantify the biotin content on the b-mNPs. B-mNPs (1 mg/mL) were diluted 1:50, 1:100, 1:200 with 1 × PBS and the standard biocytin concentration (10−60 pmol/10 μL) were freshly prepared for measuring the biotin concentration. Transmission electron microscopy (H7600; Hitachi High Technologies America) was conducted to determine the morphology of mNPs and b-mNPs.

## Optimization of processing conditions for peptide microarray

A total of 119 peptides, including 8 peptides of length 7 amino acids (aa) and 8 peptides of length 10 aa from the literature[31,32], and 103 random 15 aa peptides generated with a frequency of 5% for each of the 20 amino acids, were printed in duplicate on peptide microarrays by PEPperPRINT. The peptide microarrays contained hemagglutinin (HA) peptides (YPYDVPDYAG; 9 spots) as internal quality controls. Varying screening conditions of the peptide microarray were performed. A spectrum scan of melanin nanoparticles (mNPs) and biotinylated mNPs confirmed that the autofluorescence was near background levels after Em = 650 nm. Streptavidin DyLight 680, which was the highest wavelength (Ex = 675 nm, Em = 705 nm) that PEPperPRINT

could use in their peptide microarray system, was selected to minimize detection of melanin. Two peptide microarray copies were first pre-stained with streptavidin DyLight680 (0.2 μg/ml) and the control antibody (manufacturer: BioxCell & PEPperPrint, catalogue numbers: #RT0268, PEPperCHIP® Mouse Monoclonal anti-HA (12CA5)-DyLight800 Control; 1:2000 dilution or 0.5 μg/ml) in incubation buffer (pH 6.5 PBS with 0.005% Tween 20 and 10% Rockland blocking buffer MB-070) for 45 min at room temperature to examine background interactions and internal quality control. No background interaction of streptavidin DyLight680 or the control antibody with the 119 different peptides were observed. To screen the optimal melanin binding condition, six different washing buffers were prepared: PBS at pH 6.5 with or without 0.005% Tween 20, PBS at pH 7.4 with or without 0.005% Tween 20, and Ultra-pure water with or without 0.005% Tween 20. The Rockland blocking buffer MB-070 was used to incubate all peptide microarrays for 30 min before the melanin binding assay. Six different incubation buffers were formulated with 10% of blocking buffer in the six different washing buffers mentioned earlier. b-mNPs (10, 100, or 500 μg/ml) in six different incubation buffers were incubated with the peptide microarray for 16 h at 4 °C or room temperature. All microarrays were subsequently washed with the same type of washing buffers and incubated with 0.2 μg/mL of streptavidin DyLight680 for 45 min in the same type of incubation buffer at room temperature for detecting the b-mNPs. The peptide microarrays were then washed for 3 × 10 s with the same type of washing buffers and proceeded to quantification of spot intensity. The pilot tests suggested that 500 μg/mL of biotinylated mNPs in pH 6.5 PBS buffer at room temperature was optimal (optimal condition shown in Fig. 1d, remaining conditions shown in Supplementary Fig. 2. With the optimal flow conditions, 10 of the 16 peptides reported in the literature had detectable fluorescence intensities due to binding by b-mNPs. Quantification of spot intensities and peptide annotation were based on the 16-bit gray scale Tag Image File Format files that exhibit a higher dynamic range than the 24-bit colorized Tag Image File Format files. Microarray image analysis was done with PepSlide® Analyzer, version 1.4. The software algorithm decomposed fluorescence intensities of each spot into raw, foreground and background signal, and calculated mean median foreground intensities and spot-to-spot deviations of spot duplicates. Based on mean median foreground intensities, intensity maps were generated and interactions in the peptide maps highlighted by an intensity color code with red for high and white for low spot intensities. The PEPperPRINT protocol tolerated a maximum spot-to-spot deviation of 40%, otherwise the corresponding intensity value was zeroed. We labeled the top 20% of peptides ranked by intensities as melanin binding (23 peptides), which included 10 literature-reported peptides with non-zero fluorescent signal. The remaining peptides were labeled as non-melanin binding (96 peptides).

## Random forest classification model training with the pilot 119-peptide microarray

Random forest is an ensemble tree-based statistical machine learning model and is robust to variable noise and insensitive to variable scales[30]. Physiochemical variables and numerical representations of peptides were computed using the R packages *Peptides*, version 2.4.4[93] and *protr*, version 1.6−2[94]. The resulting 1094 variables include composition, transition, distribution, autocorrelation, conjoint triad, quasi-sequence-order descriptors, and pseudo-amino acid and amphiphilic pseudo-amino acid composition descriptors (Supplementary Data 8). The maximum value of lag was set to 6, so the minimum length of a peptide to be analyzed without generating a missing value is 7. A random forest classification model with 100,000 trees and balanced sampling was trained on the melanin binding data set. The model was built using the R package *randomForest*, version 4.7−1.1[95]. For each tree in the random forest, a bootstrap sample of -63.2% of the melanin binding peptides and the same amount of non-melanin binding

peptides was generated to construct the tree. The remaining peptides were considered out-of-bag to the tree and were used to evaluate the performance of the random forest by calculating the aggregated out-of-bag predictions across all trees. The out-of-bag class errors were calculated and a classification threshold of 0.5 proportion of votes was used. As part of the same analysis, permutation variable importance was obtained with the *importance* function in the *randomForest* package. For each tree in the random forest, out-of-bag instances were permuted for each variable in the subset, and the decrease in accuracy was recorded. The mean decrease in accuracy for each variable was calculated over all 100,000 trees and normalized by dividing the mean by the standard error.

## Expansion of the peptide microarray

Melanin binding candidate peptides were generated randomly with a frequency of 5% for each of the 20 amino acids. Peptides classified as melanin binding by the trained random forest model were selected, resulting in 5483 peptides of length ranging from 7 to 12 aa. Along with the 16 known melanin binding peptides from the literature, a total of 5499 peptides were printed in duplicate along with HA controls (YPYDVPDYAG; 68 spots) on peptide microarrays by PEPperPRINT. Peptide sequences were printed in duplicate of a custom peptide microarray. Pre-staining of a peptide microarray copy was done with streptavidin DyLight680 (0.2 µg/ml) and the control antibody (mouse monoclonal anti-HA (12CA5) DyLight800; 0.5 µg/ml) in incubation buffer to characterize non-specific binding. Subsequent incubation of another peptide microarray with the b-mNPs at a concentration of 500 µg/ml in incubation buffer (PBS at pH 6.5 with 0.005% Tween 20 with 10% Rockland blocking buffer MB-070) was followed by staining with streptavidin DyLight680 (0.5 µg/mL) and the control antibody (0.5 µg/mL). The control staining of the HA epitopes was done simultaneously as internal quality control to confirm the assay quality and the peptide microarray integrity. Quantification of spot intensities were described earlier in the previous section.

## Variable reduction of the machine learning input data

To reduce the number of variables and improve the model performance, a variable reduction procedure was applied to the machine learning input data before model training. Permutation-based variable importance was first computed on the data set with random forest (100,000 trees) using the R package *ranger*, version 0.14.1[96], with balanced sampling for classification analyses. Variables with negative importance values were removed. Next, subsets of the machine learning data set containing cumulative top-ranked variables were used to train random forests with 1000 trees, and the models were evaluated by the Akaike information criterion (AIC). The AIC values classification models were calculated using the original formula proposed by Akaike[97]: $\text{AIC} = -2\ln(\hat{L}) + 2k$, where $\hat{L}$ is the maximum likelihood value, and $k$ is the number of parameters. For regression AIC was calculated using the likelihood of normal distribution, assuming residuals are normally distributed: $\text{AIC}_{\text{reg}} = N\ln(MSE) + 2k$, where $N$ is the number of samples, and $MSE$ is the mean squared error. The classification AIC was based on the likelihood of Bernoulli distribution, and was generalized to multi-class classification: $\text{AIC}_{\text{clf}} = 2 \cdot \ln 2 \cdot N \cdot H_p(q_\theta) + 2k$, where $N$ is the number of samples, $H_p$ is denotes cross entropy, and $q_\theta$ is the estimated probability with parameters $\theta$. The variable subset with the lowest AIC value was selected for each machine learning data set.

## Machine learning model training for melanin binding predictions

Peptide variables were computed as described for the melanin binding peptides in the pilot microarray. Because the distribution of melanin binding fluorescence intensity was right-skewed, the intensity values were first normalized by $\log_{10}$-transformation for a balanced response

variable. The melanin binding data set was processed using the variable reduction method. To generate the machine learning input data set, less informative peptide variables were eliminated as described above. A nested cross-validation framework was then applied to provide an unbiased estimate of the generalization performance. The framework contains two types of cross-validations. The first includes ten sets of train-test splits computed using a Monte Carlo sampling method, which is referred to as the outer loop cross-validation. For each training set in the outer loop, another ten sets of train-test splits were generated using a modulo method. These cross-validations are referred to as the inner loop cross-validations. The inner loop cross-validations were used to select the best-performing model, and the outer loop cross-validation was used to evaluate the whole machine learning training process.

A wide array of machine learning models, including neural networks[98], gradient boosting machines (GBM)[99], extreme gradient boosting (XGBoost)[100], generalized linear model (GLM)[101], (distributed) random forests (DRF)[30], and extremely randomized trees (XRT)[102], were employed to train the input data. Hyperparameters for neural networks, GBM, and XGBoost were selected using the random grid search. Details about the grids used and the hyperparameters selected can be found in Supplementary Note 2 and the provided code. There were 300 neural networks, 300 GBM models, and 300 XGBoost models trained for the melanin binding data set, along with five default GBM models, three default XGBoost models, one GLM, one DRF, and one XRT. The model types and hyperparameters were defined based on the architecture of *H2O AutoML*[103]. For non-tree-based models, variables in the training set were scaled to have zero means and unit variances. Unstable neural networks with potentially large activation values were removed.

To integrate the models explored, a super learner (SL) model was built using the R interface of *H2O.ai*, version 3.38.0.2[104]. A generalized linear model algorithm (meta-learner) was used to calculate the coefficients (weighted contributions) of the base machine learning models according to their holdout predictions generated from the tenfold cross-validation. The meta-learner was then evaluated with another tenfold cross-validation trained on the base model holdout prediction data set. Coefficient distributions were collected from the ten cross-validation meta-learner models, resulting in a $n \times m$ matrix, where $n$ is the number of base models, and $m = 10$ is the number of cross-validation folds. The original SL algorithm used a meta-learner to calculate base model contributions and did not emphasize explicit base model selection. To reduce the complexity of SL, we developed an iterative filtering procedure to improve performance and decrease prediction run time. Specifically, base models with the number of zero coefficients >5 across cross-validation folds were removed. The filtering procedure was repeated until there were no base models or no further reduction of the base models. In addition, SL models with different compositions of base models were also constructed for comparison. Homogeneous SL ensembles were constructed with base models of the same model type (neural networks, GBM, XGBoost).

Regression models trained on the melanin binding data were evaluated in each inner loop cross-validation using multiple metrics, including coefficient of determination ($R^2$), percent normalized mean absolute error (MAE, less sensitive to outliers), and percent normalized root mean squared error (RMSE). A scoring scheme that calculates the sum of ranks of all metrics used was applied, and non-parametric Mann–Whitney U tests comparing the top model and the rest of the models were conducted to identify competitive models, with $p$ values adjusted using the Benjamini–Hochberg procedure[105]. Evaluation results regarding the competitive models whose performances were not significantly different from the top model for all evaluation metrics were reported (Supplementary Data 3). Next, the top model was selected from each inner loop cross-validation and evaluated using the

corresponding test sets in the outer loop cross-validation, and the generalization performance was computed (Supplementary Table 1).

Finally, the abovementioned model training procedure was performed on the whole data set, and the final predictive model was selected based on the same scoring scheme of the sum of all metric ranks.

## Machine learning model training for cell-penetration predictions

Cell-penetrating and non-cell-penetrating peptides of various lengths (10–61 amino acids) were collected from the *SkipCPP-Pred* website[23], for which the redundant cell-penetrating peptides from the *CPPsite2.O* database[24] have been removed, and non-cell-penetrating peptides were generated randomly[23]. There were 460 cell-penetrating and 462 non-cell-penetrating peptides. Peptide variables were computed as described above for classification of the melanin binding peptides from the pilot microarray. The variable reduction procedure as described above was then applied to the data set. A nested cross-validation framework was employed to generate train-test splits for outer and inner loop cross-validations. Multiple machine learning models were trained on the cell-penetration data set, including 100 neural network grid models, 100 GBM grid models, 100 grid XGBoost models, five default GBM models, three default XGBoost models, one DRF, and one XRT. Models were integrated using the SL framework, resulting in SL models separately containing all base machine learning models, reduced base models, all neural networks, all GBM models, and all XGBoost models. Balanced sampling was applied where appropriate for the machine learning algorithms.

Classification models were evaluated with logarithmic loss, Matthews correlation coefficient (MCC), $F_1$ (harmonic mean of precision and recall) and balanced accuracy. A scoring scheme computing the sums of all metric ranks was applied. Competitive models with no significant difference from the top model in terms of model performance, along with the means and standard errors of metrics obtained from 10-fold cross-validations were reported (Supplementary Data 6). The top model from each inner loop cross-validation was selected. The generalization performance (Supplementary Table 2) was evaluated in the outer loop cross-validation, using logarithmic loss, MCC, $F_1$, balanced accuracy, enrichment factor (EF), and Boltzmann-enhanced discrimination of receiver operating characteristic (BEDROC)[106].

The final predictive model was generated by applying the same model training procedure on the whole data set. Class prediction thresholds for the final model were selected based on the maximum $F_1$.

## Machine learning model training for cytotoxicity predictions

Toxic and non-toxic peptides of various lengths (4–35 aa) were collected from the *ToxinPred* website[25]. Peptides with length <7 were excluded, resulting in 1777 toxic and 3522 non-toxic peptides. Peptide variables were calculated as described in the random forest classification section, and non-toxic and toxic peptides were labeled as positive and negative, respectively. The dimensionality of the data set was reduced using the variable reduction as described in the above section. A nested cross-validation framework was applied, and the machine learning models include 100 neural networks, 100 GBM models, 100 XGBoost models, five default GBM models, three default XGBoost models, one GLM, one DRF, and one XRT. The mean number of peptides in non-toxic and toxic classes was calculated and used as the number of samples of each class for balanced sampling. Models were integrated using the SL framework, generating SL models containing all base models, reduced base models, all neural network models, all GBM models, and all XGBoost models. The models selected as the top model in each inner loop cross-validation were selected using the evaluation metrics and the scoring scheme as described for cell-penetration model training. Competitive models were reported in Supplementary Data 7. The generalization performance was computed

based on the selected models and recorded in Supplementary Table 3. The final predictive model was generated by performing the above model training procedure on the whole data set, and the class prediction threshold was determined by the maximum $F_1$ score.

## Peptide generation for machine learning model validation

Amino acid frequencies at each position were calculated for the 5499 melanin binding peptides used in the expanded peptide microarray, where the peptides were grouped into 8 sets based on intensity ranges. For each intensity group, random peptides were simulated based on the position-dependent amino acid frequency. In total, 127 peptides of length ranging from 7 to 12 were selected, including the TAT$_{47-57}$ peptide as the reference cell-penetrating peptide and 7 peptides from the expanded peptide microarray as validation controls. Melanin binding intensity values were predicted by the reduced melanin binding SL model. Selected peptide sequences were subsequently analyzed by the cell-penetration and toxicity final models for further classification.

## Peptide synthesis

The library of 127 C-terminal biotinylated peptides used in cell culture experiments was synthesized by Gene Script using their Crude Peptide Library service. A terminal lysine was added to each peptide sequence to facilitate biotin conjugation. Peptides from the crude peptide library were further purified by being first dissolved in 50% acetonitrile (ACN) with 0.1% TFA at 10 mg/mL. Shimadzu LC20 high-performance liquid chromatography (HPLC) system with Phenomenex reverse-phase preparative HPLC column (Gemini® 10 μm C18 110 Å, LC Column 250 × 21.2 mm, AXIA™ Packed) were used to separate and collect the peptides with an elution gradient of 5/5/90/90/5/5% solvent B (TFA 0.05% in ACN) at 0/2/10/12/13.5/15 min with a flow rate of 5 mL/min with monitoring at 220 nm.

## Melanin binding assay for machine learning model validation

The mNPs were mixed with C-terminal biotinylated peptides (10 μM) in pH 6.5 PBS solution and incubated in the rapid equilibrium dialysis (RED) 8 K device for 24 h on an orbital shaker at 900 rpm. A total of 10 μL of the solution from the rapid dialysis reservoir was collected. The concentration of unbound biotinylated peptides was analyzed with the Pierce™ Fluorescence Biotin Quantitation Kit. Four sets of melanin binding assays were performed. Melanin binding was calculated as the difference in free peptide normalized with the starting peptide concentration. Experimental melanin binding values of the 127 peptide candidates were compared with the predicted melanin binding values with the Pearson correlation. Melanin binding predictions larger than 100% were cast to 100% because this was the maximum value in the melanin binding training set.

## Cell-penetration assay with ARPE19 cell type for machine learning model validation

Three 96 well plates per ARPE19 cell type group (melanin-induced or non-melanin induced) were seeded at $0.01 × 10^6$ cells/well. ARPE-19 cells were either cultured with DMEM:F12 medium containing 10% FBS according to protocol provided by the vendor (non-melanin induced) or cultured in DMEM high glucose, pyruvate media with 250 μM of ferric ammonium citrate[107] for 2 months (melanin-induced)[20]. The expression of melanin was confirmed visually with bright field microscopy and by measuring absorbance at 475 nm (>0.4 arb. units)[20]. Within each plate, 12 wells were randomly selected to quantify the cell numbers with an automated cell counter (Countess 3 Automated Cell Counter, Thermo Fisher) for normalization in the cell uptake study. Next, 100 μL (100 μM in pH 6.5 PBS) of each of the 127 C-terminal biotinylated peptides was added to $n = 3$ wells for both the induced and non-induced ARPE-19 cells for 6 h at 37 °C. The cells were then

washed thoroughly five times with PBS solution to remove extra-cellular peptide. To quantify cell-associated peptides, the cells were lysed with 100 µL of RIPA lysis buffer at 4 °C for 48 h. The concentration of intracellular biotinylated peptides was analyzed with the Pierce™ Fluorescence Biotin Quantitation Kit. The mean intracellular concentration values of the three replicates were then grouped by cell types (melanin-induced or non-melanin induced), and a two-tailed Mann–Whitney U (Wilcoxon rank-sum) test was calculated using the *wilcox.test* function in R. The intracellular concentration values were also plotted against experimental melanin binding. The relationships between experimental cell-penetration and melanin binding values in the two ARPE19 cell type groups were quantified using the Pearson correlation.

## Shapley additive explanations (SHAP) analysis of variable contributions

To better characterize variable contributions to peptide property predictions, models trained on the outer loop training sets with the same hyperparameters as the final predictive model were used to calculate SHAP values using the corresponding test sets. For each sample in the test set, the SHAP analysis calculated the additive variable attributions to the model prediction. Specifically, models were imported using the Python interface of *H2O.ai*, version 3.38.0.2[108], and the background data set was generated by randomly selecting 100 samples from the training set. Next, SHAP values, with the number of sampling times set as 1000, were computed using the function *KernelExplainer* in the Python package *SHAP*, version 0.41.0[29]. The *KernalSHAP* method calculates variable contributions (SHAP values) using a local interpretable model-agnostic explanations (LIME) strategy[109]. The top 20 variables ranked by the difference between the maximum and minimum SHAP values in the aggregated test set samples were selected and visualized along with the variable values normalized by percentile ranks.

Explanations of HR97 multifunctional peptide predictions were computed using the final models trained on the whole machine learning data sets. The same SHAP analysis method as described above was performed, and the top variables ranked by absolute SHAP values were visualized as waterfall plots using the function *plots.waterfall* in the *SHAP* package.

## Adversarial computational controls

To assess if the model performance evaluation was overly optimistic, and if the machine learning models have learned the meaningful relationships in the data sets, adversarial controls were generated by training the models on the data sets with the response variables randomly shuffled[110]. The same nested cross-validation framework and model selection procedure as described in the above model training sections were used, and the generalization performance was computed with the models selected as the top model from the inner loop cross-validations. Statistical results of the competitive models from the inner loop cross-validations (Supplementary Note 3), and the generalization performance of the adversarial controls evaluated in the outer loop cross-validation (Supplementary Tables 1–3) were reported. Variable contributions of the adversarial control models having the same hyperparameters as the final predictive model of each property were computed and visualized as described in the SHAP analysis section.

## Peptide design space visualization

Peptide sequences of the control melanin binding peptides, pilot 119 peptides, expanded 5499 peptides, and 127 peptide candidates for experimental validation were converted using one-hot encoding, and the post-padding was applied for peptides with shorter lengths. Combined with the union set of variables from the variable-reduction

processed melanin binding, cell-penetration, and cytotoxicity data sets, the new data set was normalized and analyzed using t-Distributed Stochastic Neighbor Embedding (t-SNE), a nonlinear dimensionality reduction technique, with the *Rtsne* function in the R package *Rtsne*, version 0.16[111]. The t-SNE results were visualized along with the multi-functional predictions.

## Traceless linker system for conjugating HR97 to brimonidine

The traceless linker system was designed for release of intact parent drug when triggered by an intracellular chemical and enzymatic event, such as protease cleavage of the amide bond[112]. Activation of the linker, MC-Val-Cit-PAB-OH (Maleimidocaproyl-L-valine-L-citrul-line-p-aminobenzyl alcohol), was conducted as previously reported with minor modifications[112]. MC-Val-Cit-PAB-OH (8.68 g, 15.2 mmol) was suspended in DMF (43.4 mL) at 0 °C with water bath sonication for 30 min. After the solids were fully dispersed, thionyl chloride (1.22 mL, 16.7 mmol) was added dropwise. Following the addition, the reaction was held at 0 °C for 45 min and then treated slowly with water (130 mL) to precipitate a yellow solid (MC-Val-Cit-PAB-Cl), which was collected by filtration. The solid was washed sequentially with water and MTBE and dried under vacuum (~30% yield)[112]. Bri-monidine base was combined with the MC-Val-Cit-PAB-Cl (1.1 eq) in DMF (0.25 M) at room temperature. Tetrabutylammonium iodide (0.5 eq) was added to the solution, followed by the addition of N,N-diisopropylethylamine (2.5 eq), and the mixture was stirred for 24 h. The mixture was diluted with 50:50 acetonitrile:water at 40-fold dilution for purifying the MC-Val-Cit-PAB-brimonidine. A Shimadzu LC20 HPLC system coupled with photodiode-array detector (PDA) and with Phenomenex reverse-phase preparative HPLC column (Gemini® 10 µm C18 110 Å, LC Column 250 × 21.2 mm, AXIA™ Packed) was used to separate and collect the conjugates with an elution gradient of 10/90/90/10% solvent B (TFA 0.05% in ACN) at 1/11/13/15 min with a flow rate of 10 mL/min. The collected fractions were then transferred to the 20 mL scintillation vials and a Biotage V-10 solvent evaporator with Volatile mode was used to remove the acetonitrile. The solution fractions were frozen and lyophilized (~8% yield). NMR was used to confirm the presence of key functional groups in the products of each stage of the synthesis, including brimonidine, Mc-VC-PAB-Cl, and Mc-VC-PAB-brimonidine. All com-pounds were dissolved in deuterated DMSO and characterized with a Bruker spectrometer (500 MHz). $^{1}$H chemical shifts were reported in ppm ($\delta$) and the DMSO peak was used as an internal standard. Data were processed using TopSpin NMR Data Analysis software, version 4.1.0, from Bruker (Billerica, MA, USA). The prep-HPLC retention time (RT) of brimonidine, Mc-VC-PAB-brimonidine, and Mc-VC-PAB-Cl was 5.1, 9.8, and 11.4 min, respectively. HR97 with cysteine at the C-terminus as the functional group for linker con-jugation (FSGKRRKRKPRC, *Mw* = 1519, >97% purity) was conjugated to the quaternary-ammonium-linked brimonidine (MC-Val-Cit-PAB-brimonidine) via a thiol-maleimide reaction. The MC-Val-Cit-PAB-brimonidine was first dissolved in 1 mL of PBS at 5 mg/mL. HR97 peptide powder (0.5 eq) was added directly to the solution. The solution mixtures were adjusted to pH 7.4 and allowed to react for 2 h at room temperature. The solution mixtures were then added to 1 mL of acetonitrile and purified with the same prep-HPLC condi-tions. The collected fraction solutions were transferred to the 20 mL scintillation vials and the Biotage V-10 solvent evaporator with volatile mode were used to remove the acetonitrile. The solu-tions were lyophilized and stored at −20 °C ( ~ 35% yield). For the sample preparation and MALDI-TOF analysis, the MALDI matrix sinapic acid (10 mg) was dissolved in 1 mL of acetonitrile in water (1:1) with 0.1% TFA, and 1 µL of sample (50 µM) was deposited on the MALDI sample plate. The matrix (2 µL, 10 mg/mL) was deposited on the air-dried sample and allowed to air dry for 10–20 min. The

MALDI-TOF MS analysis was performed on a Bruker Voyager DE-STR MALDI-TOF (Mass Spectrometric and Proteomics core, Johns Hopkins University, School of Medicine) operated in linear, reflective-positive ion mode.

## In vitro melanin binding assay

Brimonidine, HR97-biotin, and HR97-brimonidine at a range of concentrations (3.125, 6.25, 12.5, 25, 50, 100 μg/mL) were dissolved in pH 6.5 PBS solution. The solutions (400 μL) were then mixed thoroughly with 400 μL of 1 mg/mL mNPs in pH 6.5 PBS solution and transferred to the inner reservoir of the rapid equilibrium dialysis (RED) device inserts (8 K MWCO). The outer reservoir was filled with 800 μL of pH 6.5 PBS solution. The samples were incubated on an orbital shaker with temperature control at 37 °C and 300 rpm for 48 h ($n = 3$). The solutions from outer reservoir (free drug) were than collected and transferred to an autosampler vial for HPLC analysis (Prominence LC2030, Shimadzu, Columbia, MD) with photodiode-array detection (PDA) system. Separation was achieved with a Luna® 5 μm C18(2) 100 Å LC column 250 × 4.6 mm (Phenomenex, Torrance, CA) at 40 °C using isocratic flow. The amount of bound drug was used to calculate the binding capacity (mol drug/mg melanin) and the dissociation constant ($K_d$) as previously described[20,34].

## In vitro stability test for HR97-brimonidine conjugate

Two pairs of human donor eyes were obtained from the Lions Gift of Sight under protocol IRB00056984 approved by the Johns Hopkins University School of Medicine Institutional Review Board. Both donors were male with an mean age of 74.5. The post-mortem times ranged from 35–40 h. The eyes were kept at 4 °C during transport and arrived within 48 h post-mortem. The vitreous and aqueous were first isolated and subsequently combined and filtered through the 0.02 μm syringe filter to remove cell debris. HR97-brimonidine (1 mg/mL) was incubated with human aqueous or vitreous (700 μL) at 37 °C ($n = 3$). On days 0, 1, 7, 14, 21 and 28, 100 μL of the solutions were collected, diluted with 900 μL of acetonitrile, and characterized by HPLC (Prominence LC2030, Shimadzu) with Luna® 5 μm C18(2) 100 Å LC column 250 × 4.6 mm (Phenomenex). The elution flow rate was 1 mL/min and with gradient of 10/90/90/10% solvent B (TFA 0.1% in ACN) in 1/11/13/15 min at $\lambda_{max} = 250$ nm for HR97-brimonidine (RT = 4.6 min). The area under the curve (AUC) on day 0 was used to normalize the AUC calculated on days 1, 7, 14, 21 and 28.

## Cathepsin cleavage assay for HR97 and HR97-brimonidine conjugate

An assay to demonstrate enzymatic cleavage of the linker was used as previously described with adaptations[112]. In brief, the HR97-brimonidine conjugate solution (200 μM) was diluted with an equal volume of 100 mM citrate buffer at pH 5.5. Cysteine was added to a final concentration of 5 mM before the addition of human cathepsins B, K, L, and S to final concentrations of 150 nM each. The mixture was then incubated for 0 h (control group) or 48 h at 37 °C. The solutions were further diluted with acetonitrile to 1 mL and conjugate concentration was measured using the HPLC method described above. All concentration values are normalized to the HR97-brimonidine at 0 h.

## Cell viability assay of HR97 peptide

The PrestoBlue™ HS cell viability system was used to assess cell viability. ARPE-19 cells were seeded at $0.01 × 10^6$ cells/well in 96 well plates and cultured with DMEM:F12 medium containing 10% FBS according to the vendor protocol. After 7 days, 90 μL of DMEM/F12 containing 0, 1, 5, 10, or 20 mg/mL of the HR97 was added. The cells ($n = 5$) were then incubated for 12 h, and viability was measured by adding 10 μL of PrestoBlue™ HS cell viability reagents at 37 °C. After 0.5, 1, 2, 3, 4 and 5 h, absorbance (570 nm and 600 nm) was measured at 37 °C and normalized according to the protocol provided by the vendor.

## Animal studies—Animal welfare statement

Experimental animal protocol (RB21M176) was approved by the Johns Hopkins Animal Care and Use Committee. All animals were handled and treated in accordance with the Association for Research in Vision and Ophthalmology Statement for Use of Animals in Ophthalmic and Vision Research. Dutch Belted rabbits (4–5 mo) were obtained from Robinson Services, Inc. Rabbit sex was uniformly distributed and randomly assigned to each group, which consisted with either 3 male/2 female, or 2 male/3 female for IOP/safety studies and 2 male/2 female for the pharmacokinetic study.

## Rabbit IOP measurements, topical dosing, and ICM injection

For the IOP measurements in normotensive rabbits, Dutch Belted rabbits (2–3 kg) were used ($n = 5$). IOP was measured with a hand-held rebound tonometer icareTONOVET (Vantaa, Finland) in the awake and gently restrained rabbit. Each rabbit was acclimatized to the IOP measurement procedure for at least 5 days to obtain a stable background IOP reading. A mean of three IOP measurements for an individual eye were taken every other day for 6 days (3 times in total) and used as a baseline value. For the ICM injection procedure, rabbits were anesthetized with ketamine/xylazine and received topical anesthesia with 0.5% proparacaine hydrochloride. A corneal pre-puncture was performed with a 30 G needle, followed with a single bolus ICM injection of 200 μg (mass of brimonidine) of HR97-brimonidine or brimonidine tartrate solution in 100 μL saline using a 28 G needle. After the procedure, topical bacitracin-neomycin-polymyxin ophthalmic ointment was applied to both eyes to prevent infection and dry eyes. On day 7, an ophthalmologist masked to treatment evaluate the HR97-brimonidine injected eyes with the following items: functionality of lids, lashes, conjunctiva, cornea transparency, pigmentation of corneal endothelium, depth of anterior chambers, inflammation, fibrin strands, and symmetry of the lens[113]. The lenses were all clear and the iris pigmentation was symmetric. In a separate study, a corneal pre-puncture was performed with a 30 G needle, followed with a single bolus ICM injection of 200 μg (mass of brimonidine) containing a physical mixture of unconjugated HR97 and brimonidine tartrate (HR97 + brimonidine), the equivalent amount of HR97 peptide alone, or saline alone in 100 μL saline. On day 7, day 14, day 21, and day 28, an ophthalmologist masked to treatment performed the same safety evaluations described above. IOP was measured on days 2, 3, 4, 5, 6, 7, 8, 10, 12, 14, 16, 18, and 20 after the ICM injection, and change in IOP from the baseline (ΔIOP) was reported. The mean of three IOP measurements was taken for each eye by one observer, and then confirmed by a masked observer. Alternatively, a single topical eye drop (Alphagan® P 0.1%, 50 μL) was given ($n = 5$). The IOP were measured immediately before the topical dosing (0 h), and at 2, 4, 6 and 8 h after the eyedrop administration. For the pharmacokinetics studies, rabbits ($n = 4$ per group) received a single ICM injection with 200 μg (mass of brimonidine) HR97-brimonidine as described above. Rabbits were sacrificed 1, 7, 14, 28 days after the injection, and iris, aqueous, and retina were collected for measuring the brimonidine concentration. One of the iris tissue samples was left out of the analysis in the day 1 group due to an issue with sample collection.

## Measurement of brimonidine in ocular tissues

Brimonidine concentrations in ocular tissues were measured by liquid chromatography-tandem mass spectrometry (LC-MS/MS) as previously described[38]. All samples were collected in pre-weighed tubes and stored at −80 °C until processing for analysis. Tissue samples were homogenized in 100–600 μL 1 × PBS using a Bullet Blender® (Next Advance, Inc, Troy, NY, USA) before extraction. Brimonidine were extracted from 15 to 50 μL of tissue homogenates with 50 μL of acetonitrile containing 50/50/2.5 ng/mL of the internal standards. The top layer was then transferred to an autosampler vial for LC-MS/MS analysis after centrifugation. All ocular tissue

samples were analyzed using a 1 × PBS standard curve for brimonidine. Separation was achieved with a Waters HSS PFP (2.1 × 50 mm, 1.8 μm). The column effluent was monitored using a 5500 mass-spectrometric detector (Sciex) using electrospray ionization operating in positive mode. The mobile phase A was water containing 0.1% formic acid and mobile phase B was acetonitrile containing 0.1% formic acid. The gradient started with mobile phase B held at 20% for 0.5 min and increased to 100% over 0.5 min; 100% mobile phase B was held for 1 min and then returned to 20% mobile phase B and allowed to equilibrate for 1 min. Total run time was 3 min with a flow rate of 0.5 mL/min. The spectrometer was programmed to monitor the following multiple reaction monitoring (MRM) transition 391.9 → 295.9 for brimonidine and 295.9 → 216.1 for the internal standard, brimonidine-d4. Calibration curve for brimonidine was computed using the area ratio peak of the analysis to the internal standard by using a quadratic equation with a x-2 weighting function over the range of 0.25–500, with dilutions of up to 1:100 (v:v). Core technicians performing sample and data analysis were masked to treatment group.

## Statistical analysis

Statistical analyses of two groups were conducted using two-tailed parametric (Student's $t$ test) or non-parametric (Mann–Whitney U) tests as appropriate. Correlation coefficients were computed using Pearson correlation (two-tailed). For multiple statistical testing, $p$ values were adjusted using the Benjamini–Hochberg procedure[105]. Statistical analyses were performed using GraphPad Prism 9 or R version 4.2.2 (2022-10-31).

## Reporting summary

Further information on research design is available in the Nature Portfolio Reporting Summary linked to this article.

## Data availability

Data sets, source data for figure generation, and all final property models are available as compressed files deposited in the Digital Repository at the University of Maryland (DRUM), https://doi.org/10.13016/0jck-hnnv[114]. Cell-penetration and cytotoxicity data sets are available on the *SkipCPP-Pred*[23] (https://bmcgenomics.biomedcentral.com/articles/10.1186/s12864-017-4128-1) and *ToxinPred*[25] (http://crdd.osdd.net/raghava/toxinpred/) websites, respectively. All other relevant data supporting the key findings of this study are available within the article and its Supplementary Information files or from the corresponding author upon reasonable request. Source data are provided with this paper.

## Code availability

The research notebook containing the code for implementing the machine learning algorithms and figure generation has been deposited in DRUM, https://doi.org/10.13016/0jck-hnnv[114]. All machine learning models can be reproduced by following the code in the research notebook in the compressed folder.

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

## Acknowledgements

We are very grateful for the veterinary and husbandry staff for their assistance, especially Sherrie Hawkes and Darlene Kyler. We thank

Carsten Haber (PEPperprint) for sharing his expertize in designing peptide microarray and reporting the outputs of the microarray. We thank Fareeha Zulfiqar for ordering and organizing the materials and equipment. We appreciate Mary Ellen Pease, Rhonda Grebe, and Arina Korneva for assisting and optimizing LSM 710 confocal microscope and TEM protocols. We acknowledge and appreciate input from Daiqin Chen regarding the traceless linker design and optimization from the original protocols. This work was supported funding to L.M.E. by the National Institutes of Health (NIH, grant nos. R01EY026578 and R01EY031041), the Robert H. Smith Family Foundation, Marcella E. Woll and the Maryland E-Nnovation Initiative Fund to establish the Marcella E. Woll Professorship in Ophthalmology, and Research to Prevent Blindness. H.T.H. was supported in part by a National Eye Institute Training Grant (T32EY007143). R.T.C. was supported in part by a National Science Foundation Award (DGE-1632976) to Michelle Girvan. Drug measurements were conducted by the Analytical Pharmacology Core (APC) of the Sidney Kimmel Comprehensive Cancer Center at Johns Hopkins, which is supported by the NIH (grant nos. P30CA006973 and S10OD020091) and the National Center for Advancing Translational Sciences (NCATS) (grant no. UL1TR001079), a component of the NIH, and the NIH Roadmap for Medical Research. The research contents are solely the responsibility of the authors and do not necessarily represent the official view of the NCATS or the NIH.

## Author contributions

H.T.H, R.T.C., I.P., U.R., W.S.L., Y.C.K., K.T.L., J.H., M.P.C., and L.M.E. designed experiments. R.T.C., H.T.H., U.R., W.S.L., M.B.A., J.P., K.T.L., C.D., P.K., A.M., H.K., M.S., N.M.A., A.H., S.V.K.R., M.E., and I.P. performed experiments and/or analyzed experimental data. R.T.C., H.T.H., W.S.L., N.M.A., K.T.L., I.P., M.P.C., and L.M.E. wrote sections of the paper. All authors read and approved the final version of the paper.

## Competing interests

H.T.H, R.T.C., J.H., M.P.C., and L.M.E. are named as inventors on the U.S. Provisional Patent Application No. 63/340,714, which covers aspects of this work, and has been jointly filed by the Johns Hopkins University and University of Maryland, College Park. The other authors declare no competing interests.
