## [Peer Review File · Nature Communications]

REVIEWER COMMENTS

Reviewer #1 (Remarks to the Author):

The investigators describe a peptide-based targeting system for sustained drug action of glaucoma drug brimonidine. They have used machine learning to find an optimal structure that binds to melanin and forms their sustained release system that prolongs drug action. A reducible linker was used to control drug release from melanin-bound peptide carriers.

Melanin has a broad spectrum of fluorescence. Does this cause problems in the peptide detection from the microarray?

Machine learning on melanin binding is a key part of the paper, even in the title. Surprisingly, the authors are missing a recent paper in which machine learning was used to find melanin binding features in a library of 3200 small molecules (J Med Chem 61: 10106-10115, 2018).

Peptide microarray is based on melanin binding. How cell permeating properties can be taken into account in the model building without data?

Cell penetration to ARPE19 is unclear. How you can differentiate between free, melanin bound and other bound peptides in the cells.

Brimonidine is applied as eye drops three times per day. This is surprising for a drug with relatively high melanin binding. Why is that? What is the required free brimonidine concentration for activity?

Prolonged action of drug has been shown previously in the case of other drugs also, not only the group's work in sunitinib. See for instance atropine and pilocarpine ((Int J Pharm 19: 53-61, 1984; J Pharmacol Exp Ther 197: 79-88, 1976).).

After intracameral injection about 20 days' duration of action was shown with a brimonidine-peptide conjugate, a relatively modest increase compared to intracameral injection of brimonidine. Intracameral injections at 20 days intervals are not an acceptable approach clinically and are much shorter than the duration of action of Allergan's implant. I wonder why the investigators did not use topical administration to compare with brimonidine eye drops. Does the peptide conjugate show too low absorption into the eye to exert good IOP responses? It is much easier to enter ciliary body from intracameral injection than from eye drops.

Reference 24 lacks bibliographic information.

Reviewer #2 (Remarks to the Author):

The presented study by Hsueh and Chou et al. describes the discovery and in vivo validation of a novel peptide-drug conjugate combining the novel peptide HR97 with the IOP-lowering drug brimonidine tartrate to create a sustained-release formulation of brimonidine for local injection. The study uses a combination of experimental and computational approaches, including a high-throughput flow-based peptide microarray and a meta (machine) learning pipeline, to create novel peptide candidates and guide their selection. The final candidate, HR97-brimonidine, reduces the intraocular pressure in rabbits for twice as long and more substantially compared to brimonidine injection and pharmacokinetic measurements indicate that brimonidine is released over multiple weeks. Overall, this is a very exciting research area and the authors have created an intriguing workflow. However, I do have multiple concerns regarding the study design and the interpretation of the results that need to be addressed to ensure better transparency and the full correctness of the here derived conclusions.

First and foremost, it appears that replicates of the machine learning model training and evaluation are missing. Many of the employed models, including deep neural networks and meta-learners, can have large variations in performance based on different cross-validation splits, heuristic training algorithms, and different random weight initializations. All model evaluations should be repeated at least ten times and average performance and standard deviation should be reported. Model performance should then be evaluated statistically to ensure that the differences observed in model performance are statistically significant – especially for the models predicting toxicity and cell penetration that have practically equivalent performances across all models (Supplementary Table 2 and 3).

Additionally, the authors need to reconsider the choice of performance metrics for their evaluations. The regression models are evaluated by RMSE and R² while the classification models are evaluated using Accuracy. Neither of these metrics are valuable evaluations in imbalanced data scenarios as presented here and more appropriate metrics for such scenarios such as F1 scores and Matthews Correlation Coefficients need to be reported. Additionally, since the goal of the machine learning model is the design of highly potent and safe peptides, more appropriate evaluation statistics such as Enrichment Factors and BEDROC analysis are required to provide insights in how the models would perform on the tasks they are used for. For example, when observing the regression performance in Figure 3b, it seems the regression model mostly performs well by correctly predicting non-binders while the performance of predicting binders is poor – the authors should provide holistic model statistics to enable clearer model interpretation and discuss such potential challenges.

Furthermore, although the peptide sequences do not seem to be provided, the two discussed peptides HR97 and TAT47-57 appear to substantially share sequence motifs and physicochemical properties. The study would tremendously benefit from a bioinformatic analysis of the peptide data to generate clusters and dimensionality reduction using measures such as Mutual Information and tools such as tSNE to better understand the complexity of the design space.

It appears the models are largely evaluated based on random cross-validation and train-test splits. Especially given the high complexity of the here proposed model and parameter optimization, I am concerned that the high performance is at least in parts explained through overfitting. Additional evaluations where the training data is split based on clusters and groups identified in the peptide sequence space is necessary to properly evaluate the generalizability of the proposed model architectures.

In addition to the above-described shortcomings of the computational analysis, the study is lacking any adversarial computational controls. Approaches such as γ -shuffling, feature ablation, and adding noise to the datasets is required to understand the robustness and correctness of the predictions and the feature importance analysis. See also ACS Chem. Biol. 2018, 13, 10, 2819–2821 for more information on such controls.

During the *in vitro* stability assay, the HR97-brimonidine seems to not be affected over a long incubation period, but it seems like a negative control is missing to quantify the normal enzymatic activity in these incubations. A known unstable peptide should be included to benchmark the degradation in the here utilized *ex vivo* samples.

For the *in vivo* validations, it is unclear why the authors did not include vehicle control (peptide injection only) and unconjugated drug/peptide mixtures. With the current experimental design, it can not be refuted that off-target effects or unspecific interactions of the designed peptide might be at least in part responsible for observed differences rather than the sustained release of the drug alone.

It appears that both the code of the machine learning algorithms and the here generated data are missing, preventing the interpretation, reproduction, and broader impact of the here created approach

and data. I would strongly recommend the authors to make all their data and code publicly available.

Finally, it is worth mentioning that the ensemble models employed here are by far the most complex model architectures that I have ever seen for similar tasks and, unfortunately, at seemingly little to no benefit compared to simpler models such as plain Random Forest architectures. While I believe it is valuable to evaluate the performance of such models, the authors should carefully discuss the benefits and challenges for such complex architectures in terms of interpretability, reproducibility, and sustainability of research. With the impressive in vivo results and for consideration in an interdisciplinary journal, it is important to be transparent about such aspects for the broader scientific community.

Reviewer #3 (Remarks to the Author):

The current work proposes an implant-free approach to sustained intraocular drug release as a means to enhance treatment adherence through a peptide-drug conjugate with high melanin binding, high cell-penetration, and low cytotoxicity. The manuscript is well-structured and supported by computational and experimental data. The authors utilized machine learning-based analyses and a high-throughput flow-based peptide microarray system to identify sequences demonstrating the aforementioned properties. Intracameral administration of the peptide-drug conjugate resulted in a sustained IOP-lowering effect for up to 20 days post administration compared to the 8-day effect of the drug and the 8-hour effect of the topically administered marketed eyedrops. This work is a significant contribution to the field of depot formulations with potential applications beyond ocular drug delivery. A few minor comments to be addressed by the authors:

1. Cell Viability assay of HR97 peptide: Why only a 12 h treatment period of the ARPE-19 cells with the HR96 peptide was chosen, when the peptide is intended to act as a drug depot for longer periods of time?
2. What is the conjugation yield?
3. Please elaborate more on the clinical application of the proposed approach. Would monthly intracameral injections of the drug conjugate be feasible/required? Would a higher dose of the drug conjugate result in a more extended IOP-lowering effect?
Based on Figure 5d, what is the timepoint (day) after which therapeutic drug concentrations are not achieved anymore.
4. Page 11-line 610 and 611: temperature is missing

REVIEWER COMMENTS

Reviewer #1:

The investigators describe a peptide-based targeting system for sustained drug action of glaucoma drug brimonidine. They have used machine learning to find an optimal structure that binds to melanin and forms their sustained release system that prolongs drug action. A reducible linker was used to control drug release from melanin-bound peptide carriers.

1.1 Melanin has a broad spectrum of fluorescence. Does this cause problems in the peptide detection from the microarray?

Yes, this is an important point. We did a spectrum scan of melanin nanoparticles (mNPs) or biotinylated mNPs and confirmed that the melanin fluorescence is the near background signal after $\text{em} = 650 \text{ nm}$. We chose DyLight 680, which is the highest wavelength ($\text{Ex} = 675 \text{ nm}$, $\text{Em} = 705 \text{ nm}$) that PEPperPRINT could use in their peptide microarray system to avoid detecting the fluorescence signal from melanin. We have added a section to the methods to clarify this important point (Line # 450–454).

1.2 Machine learning on melanin binding is a key part of the paper, even in the title. Surprisingly, the authors are missing a recent paper in which machine learning was used to find melanin binding features in a library of 3200 small molecules (J Med Chem 61: 10106-10115, 2018).

Yes, this is an important reference correlating the structural features of small molecule drugs to melanin binding. We have added the reference (number 46) to the discussion as a motivation for the work in our revised manuscript (Line # 285–287):

“Indeed, a recent study used machine learning methods to characterize the structural features of small molecule drugs that impact intrinsic melanin binding, leading to the development of a model that predicted intrinsic melanin binding with 91% accuracy⁴⁶”

1.3 Peptide microarray is based on melanin binding. How cell permeating properties can be taken into account in the model building without data?

Fortunately, there are publicly available databases containing information correlating peptide sequence to cell-penetration. We utilize the most cited databases, CPPsite2.0 and SkipCPP-Pred. As we describe in the manuscript, building our machine learning model based on the CPPsite2.0 and SkipCPP-Pred databases provided increased predication accuracy compared to what was previously described in the literature for either database.

1.4 Cell penetration to ARPE19 is unclear. How you can differentiate between free, melanin bound and other bound peptides in the cells.

The extensive washing of the cells (5 times) was aimed at maximizing the detection of intracellular vs. extracellular peptide. However, it is true that the methodology used here does not differentiate between peptide that is free or bound to melanin or other structures within the cell. The use of both non-pigmented and induced pigmented cells was aimed at determining the role of melanin binding in the cellular retention process. Indeed, there is a baseline level of peptide associated with the cells even when they are not pigmented, but there was a substantial increase in cellular localization when the cells were induced to produce melanin. We also

observed that the cells with cell-penetrating properties had increased cellular localization in both non-pigmented and pigmented cells. We have added more discussion to highlight the limitations and interpretation of the data from the in vitro ARPE-19 experiments (Line # 372–378).

1.5 Brimonidine is applied as eye drops three times per day. This is surprising for a drug with relatively high melanin binding. Why is that? What is the required free brimonidine concentration for activity?

We did observe that the binding constant of brimonidine suggested relatively high binding strength, but the melanin binding capacity was low. It has been described previously that after 2 weeks of topical eye drop use, brimonidine does accumulate in pigmented ocular tissues¹, but the unbound concentrations were much lower than the bound². Further, clearance from the aqueous and vitreous fluids was much faster than from the pigmented tissues². Together, this supports the lack of cumulative IOP lowering benefit clinically. However, we found that injecting the HR97-brimonidine conjugate not only provided a larger magnitude of IOP decrease, but the longer lasting effect (17-fold increase in area under the curve compared to free brimonidine injection). The EC50 for brimonidine in the aqueous humor has been reported as 2.9 ng/mL³. As the reviewer suggests below, injection gives greater access to the ciliary body, including the ciliary pigmented epithelium, than topical dosing.

1. Brimonidine Tartrate 0.2% W/V Eye Drops. – Summary of Product Characteristics, <https://www.medicines.org.uk/emc/product/3426/smhc>

2. Shinno, K., Kurokawa, K., Kozai, S., Kawamura, A., Inada, K., & Tokushige, H. (2017). The relationship of brimonidine concentration in vitreous body to the free concentration in retina/choroid following topical administration in pigmented rabbits. *Current Eye Research*, 42(5), 748-753.

3. Acheampong, A. A., Shackleton, M., John, B., Burke, J., Wheeler, L., & Tang-Liu, D. (2002). Distribution of brimonidine into anterior and posterior tissues of monkey, rabbit, and rat eyes. *Drug metabolism and disposition*, 30(4), 421-429.

1.6 Prolonged action of drug has been shown previously in the case of other drugs also, not only the group's work in sunitinib. See for instance atropine and pilocarpine ((Int J Pharm 19: 53-61, 1984; J Pharmacol Exp Ther 197: 79-88, 1976).).

Thank you very much for the feedback. We added details about the atropine studies to the discussion as a motivation (Line # 280–282).

“In the case of atropine, the intrinsic melanin binding properties were shown to lead to prolonged residence time in pigmented rabbit ocular tissues⁴⁴, and a sustained miotic response in pigmented rabbits⁴⁵.”

1.7 After intracameral injection about 20 days' duration of action was shown with a brimonidine-peptide conjugate, a relatively modest increase compared to intracameral injection of brimonidine. Intracameral injections at 20 days intervals are not an acceptable approach clinically and are much shorter than the duration of action of Allergan's implant. I wonder why the investigators did not use topical administration to compare with brimonidine eye drops. Does the peptide conjugate show too low absorption into the eye to exert good IOP responses? It is much easier to enter ciliary body from intracameral injection than from eye drops.

We totally agree that intracameral injections at 20 day intervals would not be clinically acceptable, and we have added to the discussion to highlight this limitation. We used brimonidine for the proof-of-principle study because of the potential for observing measurable IOP lowering in normotensive rabbits. Bimatoprost is a much more potent drug (clinical eye drop formulation of bimatoprost (Lumigan) is 0.01% vs Alphagan 0.1% for brimonidine), but does not reproducibly lower IOP in many animal species, including rabbits. Our plan for future work is to focus on more potent drugs for peptide conjugation to increase the duration of action. Given the issues of adherence to eye drops and the movement of the field toward sustained delivery products like Durysta, we focused on the injection route of administration. Though we do agree that ultimately developing an eye drop that could be used intermittently and achieve sustained effects without the need for injection would also be a potentially impactful innovation.

1.8 Reference 24 lacks bibliographic information.

Thank you, we have updated the reference accordingly.

Reviewer #2:

The presented study by Hsueh and Chou et al. describes the discovery and in vivo validation of a novel peptide-drug conjugate combining the novel peptide HR97 with the IOP-lowering drug brimonidine tartrate to create a sustained-release formulation of brimonidine for local injection. The study uses a combination of experimental and computational approaches, including a high-throughput flow-based peptide microarray and a meta (machine) learning pipeline, to create novel peptide candidates and guide their selection. The final candidate, HR97-brimonidine, reduces the intraocular pressure in rabbits for twice as long and more substantially compared to brimonidine injection and pharmacokinetic measurements indicate that brimonidine is released over multiple weeks. Overall, this is a very exciting research area and the authors have created an intriguing workflow. However, I do have multiple concerns regarding the study design and the interpretation of the results that need to be addressed to ensure better transparency and the full correctness of the here derived conclusions.

We greatly appreciate the support and the constructive feedback.

2.1 First and foremost, it appears that replicates of the machine learning model training and evaluation are missing. Many of the employed models, including deep neural networks and meta-learners, can have large variations in performance based on different cross-validation splits, heuristic training algorithms, and different random weight initializations. All model evaluations should be repeated at least ten times and average performance and standard deviation should be reported. Model performance should then be evaluated statistically to ensure that the differences observed in model performance are statistically significant – especially for the models predicting toxicity and cell penetration that have practically equivalent performances across all models (Supplementary Table 2 and 3).

We thank the reviewer for the valuable comments. We previously performed 10-fold cross-validation for each model but reported the aggregated results rather than calculating the means and standard errors. We have combined your suggestion from comment 2.4 and employed a nested cross-validation framework involving model selection and evaluation. The use of nested cross-validation is to avoid potential overfitting caused by model selection. The nested cross-validation had 10 sets of train-test splits generated using a Monte Carlo method in the outer loop and 10 sets of train-test splits generated using a modulo method in each inner loop. The

inner loop cross-validations were used to select the best-performing models, and the outer loop cross-validation was used to evaluate the whole pipeline.

We have identified multiple competitive models with non-parametric Mann–Whitney U tests in each inner loop cross-validation and developed a scoring scheme that calculates the sum of the ranks of all metrics used (for regression these comprise mean absolute error, root mean squared error, and coefficient of determination or R^2 ; for classification these comprise logarithmic loss, Matthews correlation coefficient, F_1 , and accuracy). The best-performing models in each loop were selected with the smallest sum of ranks across the metrics. The p -values were adjusted using the Benjamini–Hochberg procedure. We reported those competitive models whose performances were not significantly different from the top one model, along with their means and standard errors evaluated by multiple metrics in Supplementary Data 3, 6, 7.

2.2 Additionally, the authors need to reconsider the choice of performance metrics for their evaluations. The regression models are evaluated by RMSE and R2 while the classification models are evaluated using Accuracy. Neither of these metrics are valuable evaluations in imbalanced data scenarios as presented here and more appropriate metrics for such scenarios such as F1 scores and Matthews Correlation Coefficients need to be reported. Additionally, since the goal of the machine learning model is the design of highly potent and safe peptides, more appropriate evaluation statistics such as Enrichment Factors and BEDROC analysis are required to provide insights in how the models would perform on the tasks they are used for. For example, when observing the regression performance in Figure 3b, it seems the regression model mostly performs well by correctly predicting non-binders while the performance of predicting binders is poor – the authors should provide holistic model statistics to enable clearer model interpretation and discuss such potential challenges.

We thank the reviewer for the constructive feedback. We have retrained our regression and classification models using a nested cross-validation framework and evaluated the models with the metrics for imbalanced data as suggested. Specifically, for melanin binding regression, because the distribution of the response variable was right-skewed due to the property of fluorescence intensities, we log-transformed the distribution first before model training. We then evaluated the regression models with mean absolute error, root mean squared error, and coefficient of determination. The mean absolute error was chosen because it is insensitive to outliers and thus provides a more unbiased evaluation for regression data with an asymmetric response variable. For cell-penetration and cytotoxicity classification models, we addressed the data imbalance issue with balanced sampling before fitting the data to the model. Regarding the model performance, we reported logarithmic loss, Matthews correlation coefficient (MCC), F_1 , and accuracy. The logarithmic loss evaluates the probability, MCC is good for imbalanced data, F_1 focuses on positive class prediction, and accuracy is highly interpretable.

We have evaluated the enrichment factor (EF) and BEDROC. These two metrics are used in the virtual screening context, which emphasizes the ability of a computational or machine learning model to identify highly potent molecules from the database. This differs from our approach in designing a multifunctional peptide, where we probabilistically generated peptides and selected peptide candidates based on predictions rather than employing a model to identify potential peptides. The main difference is that our models are used to classify the desired properties, and the virtual screening models are used to identify the peptides based on predicted attributes. Thus, the EF and BEDROC did not provide much information during our model selection process. However, we still included results from the two metrics for reference, which can be found in the compressed Supplementary Data folder and the Bookdown research notebook. We have also reported all six metrics (logarithmic loss, MCC, F_1 , accuracy, EF, and BEDROC) for

the final, unbiased evaluation of the machine learning pipeline with the nested cross-validation framework (Supplementary Tables 1–3).

The observation of poor prediction of higher melanin binding levels is due to the right-skewed distribution of fluorescence intensity. We have corrected this phenomenon by log-transforming the intensity values and updated Figure 3b in the manuscript. The figure shows a high correlation between the predicted and experimentally validated values. We further provided the Pearson correlation coefficient, the p -value, and the 95% confidence interval.

2.3 Furthermore, although the peptide sequences do not seem to be provided, the two discussed peptides HR97 and TAT47-57 appear to substantially share sequence motifs and physicochemical properties. The study would tremendously benefit from a bioinformatic analysis of the peptide data to generate clusters and dimensionality reduction using measures such as Mutual Information and tools such as tSNE to better understand the complexity of the design space.

We thank the reviewer for the valuable suggestion. We have included the sequence information in Supplementary Data 1, 2, 4, 5. We also performed an additional analysis with t-SNE plots (Fig. 5), which improved our understanding of peptide relationships in the design space. Specifically, the design space was defined by combining the one-hot encoded sequences and variables used in model training, representing sequence motifs and relevant physicochemical properties in melanin binding, cell-penetration, and cytotoxicity. Indeed, HR97 and TAT₄₇₋₅₇ showed up in the same cluster, sharing similar sequence motifs and peptide variables. The cluster corresponded to high melanin binding, cell-penetration, and non-toxic predictions. Both HR97 and TAT₄₇₋₅₇ consist of multiple arginine and lysine amino acids, which supports the model interpretation analysis showing that higher net charge and more basic amino acids in the sequences may contribute to higher melanin binding. HR97 was selected based on the optimal combination of *in vitro* melanin binding value ($melanin\ binding_{HR97} = 79.1 \pm 0.7\%$, $melanin\ binding_{TAT_{47-57}} = 77.6 \pm 1.5\%$), *in vitro* cell-penetration value ($cell\ uptake_{HR97} = 759.9 \pm 19.6\text{ pmol/100K cells}$, $cell\ uptake_{TAT_{47-57}} = 457.1 \pm 34.2\text{ pmol/100K cells}$), and non-toxic prediction ($Prob(non-toxic_{HR97}) = 96.9\%$, $Prob(non-toxic_{TAT_{47-57}}) = 97.5\%$).

2.4 It appears the models are largely evaluated based on random cross-validation and train-test splits. Especially given the high complexity of the here proposed model and parameter optimization, I am concerned that the high performance is at least in parts explained through overfitting. Additional evaluations where the training data is split based on clusters and groups identified in the peptide sequence space is necessary to properly evaluate the generalizability of the proposed model architectures.

We appreciate the reviewer's suggestion. Combining the suggestion from comment 2.1, we exploited a nested cross-validation to calculate an unbiased estimate of the generalization performance of our proposed model architectures, where we had comprehensively explored as many ML models as possible in this study. To consistently estimate the generalization performance throughout, we performed a Monte Carlo method to generate 10 sets of train-test splits for the outer cross-validation for the melanin binding, cell-penetration, and cytotoxicity data sets. The evaluation results have been reported in the manuscript (Line # 135–142, 151–155, and 159–162).

Regarding this comment and others, it is fundamentally important to recognize that the machine learning analyses are not done in isolation, and that we are multiple laboratory assays that

establish the functional properties of the best and other peptides. Although our machine learning analyses employ numerous mechanisms to measure performance, prevent overfitting, and address other possible concerns, we ultimately rely on empirical validation.

2.5 In addition to the above-described shortcomings of the computational analysis, the study is lacking any adversarial computational controls. Approaches such as y-shuffling, feature ablation, and adding noise to the datasets is required to understand the robustness and correctness of the predictions and the feature importance analysis. See also ACS Chem. Biol. 2018, 13, 10, 2819–2821 for more information on such controls.

Following the reviewer's recommendation, we have trained adversarial control models for melanin binding, cell-penetration, and cytotoxicity data sets using the same model architectures with the data sets having the response variable randomly shuffled. A nested cross-validation method was employed to evaluate the overall machine learning performance and model interpretation of the control models. We provided the evaluation results in Supplementary Tables 1–3 and model interpretation results in Supplementary Figure 7. Indeed, the adversarial controls had poor performance compared to their general counterparts for all three data sets, demonstrating that the ML models had learned the real relationships between the predictor and response variables. Comparing the SHAP analysis results, the distributions and the levels of variable contributions have also changed for all three adversarial control models.

2.6 During the in vitro stability assay, the HR97-brimonidine seems to not be affected over a long incubation period, but it seems like a negative control is missing to quantify the normal enzymatic activity in these incubations. A known unstable peptide should be included to benchmark the degradation in the here utilized ex vivo samples.

Thank you for addressing this point. Generally, cathepsin enzymes are located intracellularly to aid in the cellular recycling of macromolecules^{4,5,6}. In one of the most comprehensive studies of ocular distribution of cathepsins, cathepsins were found in the lysosomes in various cornea, ciliary body, iris, and retinal cells⁵. However, cathepsins are not present in appreciable amounts in ocular fluids, such as the aqueous humor and vitreous⁷, which is consistent with the observed stability of the conjugate. We have added a section to the discussion to clarify this important point.

4. Appelqvist, H., Wäster, P., Kågedal, K., & Öllinger, K. (2013). The lysosome: from waste bag to potential therapeutic target. *Journal of molecular cell biology*, 5(4), 214-226.

5. Wassélius, J., Wallin, H., Abrahamson, M., & Ehinger, B. (2003). Cathepsin B in the rat eye. *Graefe's archive for clinical and experimental ophthalmology*, 241, 934-942.

6. Rakoczy, P. E., Sarks, S. H., Daw, N., & Constable, I. J. (1999). Distribution of cathepsin D in human eyes with or without age-related maculopathy. *Experimental eye research*, 69(4), 367-374.

7. Goel, M., Picciani, R. G., Lee, R. K., & Bhattacharya, S. K. (2010). Aqueous humor dynamics: a review. *The open ophthalmology journal*, 4, 52.

2.7 For the in vivo validations, it is unclear why the authors did not include vehicle control (peptide injection only) and unconjugated drug/peptide mixtures. With the current experimental design, it can not be refuted that off-target effects or unspecific interactions of the designed peptide might be at least in part responsible for observed differences rather than the sustained release of the drug alone.

Thank you for bringing up this valuable point. We have added three control groups for the normotensive rabbit intracameral injections ($n = 5$ per group) as Supplementary table 4-7. The groups include saline injection only, peptide (HR97) injection only, and the mixture of unconjugated brimonidine + HR97. Both HR97 and saline shared the same IOP lowering profile, suggesting the peptide itself has no effect on IOP. The saline control also demonstrated that there was a transient reduction in IOP due to the intracameral injection, which returned to baseline within 3 days for both the saline and HR97 groups. In addition, the physical mixture of brimonidine and HR97 had a similar IOP profile to the brimonidine injection alone. These experimental groups add confidence to the sustained IOP lowering being provided by the conjugation of brimonidine to HR97 and binding to melanin.

2.8 It appears that both the code of the machine learning algorithms and the here generated data are missing, preventing the interpretation, reproduction, and broader impact of the here created approach and data. I would strongly recommend the authors to make all their data and code publicly available.

We thank the reviewer for the helpful suggestion. We organized all data and code in a compressed Supplementary Data folder deposited in the Digital Repository at the University of Maryland (DRUM) with the identifier <https://doi.org/10.13016/0jck-hnnv>. The folder includes a research notebook generated with R Bookdown, which contains the code of the machine learning pipeline, model training, model interpretation, experimental validation of the predictions, and figure generation. For algorithms involving random sampling, random seeds are also provided for reproducibility. The notebook also offers a small data set demo to show how to run the main machine learning pipeline, which includes implementations of the proposed variables reduction procedure and base model reduction for super learners. The code has been tested under R version 4.2.2 (2022-10-31) and Python version 3.8.15, with RStudio version 2022.12.0+353, on macOS Big Sur version 10.16.

2.9 Finally, it is worth mentioning that the ensemble models employed here are by far the most complex model architectures that I have ever seen for similar tasks and, unfortunately, at seemingly little to no benefit compared to simpler models such as plain Random Forest architectures. While I believe it is valuable to evaluate the performance of such models, the authors should carefully discuss the benefits and challenges for such complex architectures in terms of interpretability, reproducibility, and sustainability of research. With the impressive in vivo results and for consideration in an interdisciplinary journal, it is important to be transparent about such aspects for the broader scientific community.

We thank the reviewer for bringing up this important point. This study comprehensively explored a wide array of possible machine learning models with various hyperparameters for complex biological data sets. Indeed, with the statistical analyses for selecting the final models trained on the whole data sets, Supplementary Data 3, 6, 7 showed that there were 31, 300, and 175 competitive models (including super learners) for melanin binding, cell-penetration, and cytotoxicity data sets, respectively. Based on the scoring scheme of the sum of the metric ranks, the final predictive models selected for the three peptide properties were: a final reduced complexity super learner for melanin binding, a GBM model for cell-penetration, and a final reduced complexity super learner for cytotoxicity.

Although not all final models are super learners, the inclusion of a super learner framework in this study was based on the mathematical theory that the performance of a super learner will be at least as best as the best performing based model⁸. It was intended to integrate the machine learning models we have explored. The meta-learner, a generalized linear model, may add a

layer of complexity. However, it provided an interpretable summary of the base models' importance in contributions to the final super learner predictions.

We have also reduced the complexity of the machine learning analysis with two strategies. First, before training the machine learning models, we conducted a variable reduction procedure with the Akaike information criterion (AIC) to eliminate irrelevant and less informative variables. Secondly, we reduced the complexity of the super learner by iteratively removing base models with fewer contributions to the final predictions. We believe that the reduction of complexity in both machine learning data sets and super learner models helped improve the reproducibility and sustainability of the research, and have discussed the machine learning complexity in the updated manuscript (Line # 359–364).

8. Van der Laan, M. J., Polley, E. C., & Hubbard, A. E. (2007). Super learner. *Statistical applications in genetics and molecular biology*, 6(1).

Reviewer #3

The current work proposes an implant-free approach to sustained intraocular drug release as a means to enhance treatment adherence through a peptide-drug conjugate with high melanin binding, high cell-penetration, and low cytotoxicity. The manuscript is well-structured and supported by computational and experimental data. The authors utilized machine learning-based analyses and a high-throughput flow-based peptide microarray system to identify sequences demonstrating the aforementioned properties. Intracameral administration of the peptide-drug conjugate resulted in a sustained IOP-lowering effect for up to 20 days post administration compared to the 8-day effect of the drug and the 8-hour effect of the topically administered marketed eyedrops. This work is a significant contribution to the field of depot formulations with potential applications beyond ocular drug delivery. A few minor comments to be addressed by the authors:

We appreciate the support and the constructive feedback.

3.1 Cell Viability assay of HR97 peptide: Why only a 12 h treatment period of the ARPE-19 cells with the HR96 peptide was chosen, when the peptide is intended to act as a drug depot for longer periods of time?

Thank you for bringing up this valuable point. The 12 h treatment period was indicated by the manufacturer's instructions for the assay, and does evaluate acute toxicity. To address this concern, we incorporated a safety study in the additional rabbit IOP experiment described above to address the additional controls (saline, HR97, HR97 mixed with brimonidine) (Supplementary Tables 4–7). On day 7, 14, 21 and 28, an ophthalmologist masked to treatment found no evidence of intraocular toxicity with any of the intracameral treatments. A needle track was noted that resolved by day 28.

3.2 What is the conjugation yield?

The overall conjugation yield was low, and we have added to the discussion to address this limitation. We have also added the following information regarding the yield at each step of the reaction to the methods: The yield of activating MC-Val-Cit-PAB-OH to MC-Val-Cit-PAB-Cl was ~30%, conjugating brimonidine to MC-Val-Cit-PAB-Cl gave at most 8% yield after prep-HPLC purification, and conjugating the MC-Val-Cit-PAB-brimonidine to HR97 yielded approximately 35% after purification. While we attempting to increase the efficiency of the reactions at each

step, we were unsuccessful. For future work, the linker release rate should be further optimized as well, which we have also mentioned in the paragraph on limitations in the discussion.

3.3 Please elaborate more on the clinical application of the proposed approach. Would monthly intracameral injections of the drug conjugate be feasible/required? Would a higher dose of the drug conjugate result in a more extended IOP-lowering effect?

Based on Figure 5d, what is the timepoint (day) after which therapeutic drug concentrations are not achieved anymore.

As described above, we agree that more work is needed to develop conjugates with even longer duration of action to be more clinically relevant. Agreeing with your logic, we did inject the HR97-brimonidine conjugate near the solubility limit to try to maximize the duration of effect. As described above, we anticipate that the use of more potent drugs will be helpful in this regard, and we are currently pursuing this as next steps. Based on the pharmacokinetic data in Figure 5f and the comparisons to what we measured previously after eye drop dosing (dotted and dashed lines), the aqueous concentrations appear to decrease below the therapeutic concentration between days 14-28, which is consistent with the trajectory of the IOP lowering shown in Fig 6d. It was notable that the level of brimonidine bound to the iris was still quite high, suggesting that improving the linker cleavage and brimonidine release rate may also extend the duration of therapeutic effect. We had added this point to the paragraph on limitations in the discussion.

3.4 Page 11-line 610 and 611: temperature is missing

We have added the temperature symbol back in the manuscript.

REVIEWER COMMENTS

Reviewer #1 (Remarks to the Author):

The authors have answered my criticism adequately.

Reviewer #2 (Remarks to the Author):

The manuscript and study has significantly improved based on the responsiveness of the authors to all reviewers. In particular, the authors have adequately addressed all my concerns by including important control calculations and experiments and by transparently sharing their data and code.

I have a few additional minor comments in response to the reviewers changes.

1) The authors have now included a set of clearer performance metrics in their model evaluation, including MCC and F1 score. However, the inclusion of accuracy as an "interpretable" metric can still be misleading since it can overemphasize model performance in imbalanced data. Instead, balanced accuracy can provide a more realistic performance evaluation while maintaining interpretability. I recommend to replace accuracy through balanced accuracy.

2) The nested cross-validation improves the statistical analysis of the models, but does not address my concern about model generalizability. While I agree with the authors that prospective tests, as performed here, are the most powerful model evaluations, they can be insufficient to characterize the model performance when new designs are tested that are highly similar to the training data. As the authors agree in their response and show in the new Figure 5, HR97 and TAT47-57 cluster in sequence space in a region that is considered uniformly beneficial by the model. The reader is left wondering whether machine learning is necessary to select novel peptides, or whether simply maximizing sequence similarity and/or maximizing properties such as net charge could have led to the same candidate selection and impressive in vivo results. The manuscript would benefit from highlighting "surprising" negative results that are correctly identified by the model or by evaluating model performance while not randomly splitting into train/test sets but instead split along sequence properties to assess the model's ability to generalize beyond sequence and property similarity. This would allow the authors to further highlight the true predictive power of the model.

3) Finally, it appears the hyperlink to the code repository is currently broken in the manuscript since it includes the reference number "113".

Reviewer #3 (Remarks to the Author):

Authors have adequately addressed reviewer's comments and recognized the areas of their approach that would benefit from further exploration. Recommendation: Accept in current form.

REVIEWER COMMENTS

Reviewer #1:

The authors have answered my criticism adequately.

Reviewer #2:

The manuscript and study has significantly improved based on the responsiveness of the authors to all reviewers. In particular, the authors have adequately addressed all my concerns by including important control calculations and experiments and by transparently sharing their data and code.

I have a few additional minor comments in response to the reviewers changes.

2.1 The authors have now included a set of clearer performance metrics in their model evaluation, including MCC and F1 score. However, the inclusion of accuracy as an "interpretable" metric can still be misleading since it can overemphasize model performance in imbalanced data. Instead, balanced accuracy can provide a more realistic performance evaluation while maintaining interpretability. I recommend to replace accuracy through balanced accuracy.

Thank you for the valuable suggestion. We conducted our analyses with balanced sampling. To make these results clearer, we have replaced the accuracy with balanced accuracy and updated the results in the manuscript (lines 153–155 and 160–162), Supplementary Fig. 4, Supplementary Tables 2, 3, and Supplementary Data 6, 7.

2.2 The nested cross-validation improves the statistical analysis of the models, but does not address my concern about model generalizability. While I agree with the authors that prospective tests, as performed here, are the most powerful model evaluations, they can be insufficient to characterize the model performance when new designs are tested that are highly similar to the training data. As the authors agree in their response and show in the new Figure 5, HR97 and TAT47-57 cluster in sequence space in a region that is considered uniformly beneficial by the model. The reader is left wondering whether machine learning is necessary to select novel peptides, or whether simply maximizing sequence similarity and/or maximizing properties such as net charge could have led to the same candidate selection and impressive in vivo results. The manuscript would benefit from highlighting "surprising" negative results that are correctly identified by the model or by evaluating model performance while not randomly splitting into train/test sets but instead split along sequence properties to assess the model's ability to generalize beyond sequence and property similarity. This would allow the authors to further highlight the true predictive power of the model.

We thank the reviewer for the constructive feedback. The research aim is to design multifunctional peptides with simultaneously high melanin binding, high cell-penetration, and low cytotoxicity. As we can see in the interpretation of the final peptide property models (Fig. 4a,b and Supplementary Fig. 6), these different desired properties had different sets of important variables that contributed more to model predictions, and there was no single variable dominating the final predictions across all properties. We recognize, sequence similarity and properties are not independent. Thus, we agree increasing sequence similarity to a performant peptide is one path to obtain additional peptides with desired properties. However, it is also the

case that variable combinations favoring high melanin binding leads to high cytotoxicity (Fig. 4c).

As a counter example of sequence similarity, two peptides from our laboratory-validated set (t-SNE 1: -2.2, t-SNE 2: -20.5 and t-SNE 1: 43.4, t-SNE 2: -45.7) simultaneously have the three desired properties and are neither similar to HR97 nor TAT₄₇₋₅₇.

2.3 Finally, it appears the hyperlink to the code repository is currently broken in the manuscript since it includes the reference number "113".

Thank you for pointing out this issue. We identified that the problem was caused by the Word-to-PDF conversion in the manuscript submission system.

Reviewer #3

Authors have adequately addressed reviewer's comments and recognized the areas of their approach that would benefit from further exploration. Recommendation: Accept in current form.

REVIEWERS' COMMENTS

Reviewer #2 (Remarks to the Author):

The authors have adequately addressed all my concerns and I would like to thank the authors for being responsive to all reviewers and congratulate them on this excellent study.

REVIEWER COMMENTS

Reviewer #2:

The authors have adequately addressed all my concerns and I would like to thank the authors for being responsive to all reviewers and congratulate them on this excellent study.

We very much appreciate the support and valuable comments from the reviewer as well.